**DOI: 10.1038/ncomms14272**　**OPEN**

# Delaying histone deacetylase response to injury accelerates conversion into repair Schwann cells and nerve regeneration

Valérie Brügger[1], Mert Duman[1], Maëlle Bochud[1], Emmanuelle Münger[1], Manfred Heller[2], Sophie Ruff[1] & Claire Jacob[1]

The peripheral nervous system (PNS) regenerates after injury. However, regeneration is often compromised in the case of large lesions, and the speed of axon reconnection to their target is critical for successful functional recovery. After injury, mature Schwann cells (SCs) convert into repair cells that foster axonal regrowth, and redifferentiate to rebuild myelin. These processes require the regulation of several transcription factors, but the driving mechanisms remain partially understood. Here we identify an early response to nerve injury controlled by histone deacetylase 2 (HDAC2), which coordinates the action of other chromatin-remodelling enzymes to induce the upregulation of Oct6, a key transcription factor for SC development. Inactivating this mechanism using mouse genetics allows earlier conversion into repair cells and leads to faster axonal regrowth, but impairs remyelination. Consistently, short-term HDAC1/2 inhibitor treatment early after lesion accelerates functional recovery and enhances regeneration, thereby identifying a new therapeutic strategy to improve PNS regeneration after lesion.

[1] Department of Biology, University of Fribourg, Chemin du Musée 10, 1700 Fribourg, Switzerland. [2] Proteomics and Mass Spectrometry Core Facility, Department of Clinical Research, University of Bern, Freiburgstrasse 15, 3010 Bern, Switzerland. Correspondence and requests for materials should be addressed to C.J. (email: claire.jacob@unifr.ch).

Axons of the peripheral nervous system (PNS) have a high capacity of regeneration after lesion, in contrast to axons of the central nervous system (CNS), which poorly regenerate. This is due to intrinsic regenerative properties of PNS neurons, and to a large extent to extrinsic factors that allow and promote axonal regeneration in the PNS[1]. Schwann cells (SCs), the PNS myelinating glia, hold major functions in creating a favourable environment for axonal regrowth, stimulating axon outgrowth after lesion, and rebuilding myelin sheaths of regenerated axons[2]. Upon lesion, mature SCs convert into a repair cell phenotype that resembles the immature SC stage in some but not all aspects[1–3]. Indeed, repair SCs downregulate myelin proteins and pro-myelinating factors such as Krox20 to dedifferentiate and demyelinate, but they also simultaneously activate a repair programme that promotes axonal regrowth and survival, as well as axon debris and myelin removal[4]. The transcription factor cJun plays a central role in controlling these processes: cJun is strongly upregulated in SCs after a PNS lesion where it induces SC dedifferentiation, the production of neurotrophic and axon survival factors such as GDNF and Artemin, and myelin clearance by SC myelinophagy[5–8]. Other myelination inhibitors including Sox2, Pax3, Notch, Id2 are also re-expressed in SCs after lesion and are thought to participate in the SC dedifferentiation process[4]. Once converted into repair cells, SCs proliferate and migrate along damaged axons to organize into bands of Bungner that stimulate axonal regrowth and guide axons back to their former peripheral target. When axons have regrown, SCs downregulate myelination inhibitors and upregulate Krox20 to induce remyelination together with the major transcription factor of SC differentiation Sox10 (ref. 9). In contrast to Krox20, the intermediate inducer of SC differentiation Oct6 is upregulated after lesion and downregulated as SCs redifferentiate[10–12]. Oct6 is a key transcription factor for PNS development and regeneration, allowing timely myelination and remyelination by inducing Krox20 expression[13–15], but needs to be downregulated for myelination to proceed[16]. Oct6 thus importantly participates in triggering the SC differentiation programme, but also maintains SCs in a pre-myelinating stage.

In summary, SC plasticity after lesion requires dynamic regulation of several sets of transcription factors, some inducing SC dedifferentiation or conversion into repair cells, and some triggering SC redifferentiation and remyelination. Mechanisms controlling the regulation of these transcription factors are partially understood. In this study, we set out to elucidate the mechanisms controlling SC conversion into repair cells and redifferentiation after lesion with a focus on chromatin-remodelling events.

We previously showed that the chromatin-remodelling enzymes histone deacetylase (HDAC)1 and HDAC2 are essential for the specification of neural crest cells into peripheral glia[17], for SC survival and myelination during postnatal development[18] (also shown by Chen et al.[19]), and for the maintenance of PNS integrity in adults[20]. While histone acetyltransferases (HATs) add acetyl groups to histone tails, HDACs remove these acetyl groups. Because acetyl groups neutralize the positive charges of histones, histone acetylation loosens the attraction of negatively charged DNA to histones and thus leads to a more relaxed chromatin structure. In contrast, HDACs allow histones to recover their positive charges, which leads to their tight interaction with DNA, and thus to a more compacted chromatin structure. Because chromatin compaction limits the access for the transcriptional machinery to DNA, HDACs have commonly been thought to act as transcriptional repressors[21]. However, an increasing number of studies from independent groups reveal that HDACs can also participate in transcriptional activation[22–24]. Furthermore, HDACs can deacetylate non-histone targets to modify their activity[25]. Among those, several transcription factors have been described. HDACs are thus very potent transcriptional regulators and their action is highly dependent on their binding partners. Indeed, HDACs cannot bind chromatin directly, but needs to associate with DNA-binding proteins to regulate gene transcription. HDACs are known to belong to different chromatin-remodelling complexes, which often also comprise other chromatin-remodelling enzymes such as histone methyltransferases (HMTs) and demethylases (HDMs)[26]. In this study, we show that HDAC2 interacts with the transcription factor Sox10 and recruits histone H3 lysine 9 (H3K9) HDMs to form a multifunctional protein complex that de-represses the Sox10 target genes Oct6 and Krox20 to allow their subsequent activation at different time points of the regeneration process after lesion. Interestingly, inactivating this mechanism results in earlier conversion into repair SCs after lesion and faster regeneration, but impairs remyelination.

## Results

**HDAC1/2 slow down axonal regrowth but promote remyelination.** There are eighteen known mammalian HDACs, subdivided into four different classes, based on their structure. HDAC1 and HDAC2 (HDAC1/2) are two highly homologous nuclear class I HDACs that can efficiently compensate for the loss of each other[17–20]. Here, we found that HDAC1/2 were robustly regulated after sciatic nerve crush lesion in adult mice. HDAC2 was upregulated at 1 day post lesion (dpl) and remained highly expressed in SCs until completion of the regeneration process (Fig. 1a,b). Interestingly, HDAC2 was SUMOylated in adult nerves (Fig. 1c) and SUMOylation was increased after lesion (Fig. 1a), suggesting modulation of HDAC2 activity, binding partners and/or stability[27] after lesion. HDAC1 was also upregulated in SCs, but later, starting from 3 dpl (Fig. 1a,b). HDAC1/2 upregulation and HDAC2 SUMOylation suggested important functions of these HDACs after lesion. To identify these potential functions and avoid compensatory mechanisms between HDAC1 and HDAC2, we ablated both HDACs in adult SCs by crossing tamoxifen-inducible P0CreERT2 (ref. 28) with floxed Hdac1/Hdac2 mice[29]. We lesioned sciatic nerves of double homozygous knockout mice (dKO) and their controls at 7 days post tamoxifen injections, when HDAC1/2 are efficiently lost[20], including the SUMOylated form of HDAC2 (Supplementary Fig. 1). We analysed lesioned sciatic nerves at different time points after lesion and also unlesioned contralateral sciatic nerves as internal control for each mouse. We found by bromodeoxyuridine (BrDU) incorporation that proliferation was induced earlier after lesion in dKO compared with controls nerves (Fig. 2a), while we did not detect BrDU-positive cells in contralateral dKO or control nerves (Supplementary Fig. 2a). At 3dpl, all BrDU-positive cells were negative for the macrophage marker F4/80 (Supplementary Fig. 2b), indicating that proliferating cells were most likely SCs at this time point. Earlier proliferation suggested earlier SC conversion into repair cells in the absence of HDAC1/2. Consistently, demyelination occurred earlier in dKO nerves, as shown by a decreased percentage of intact myelin rings at 5 dpl (Fig. 2b). Repair SCs promote axonal regrowth[1–3,30]. Indeed, at 3 dpl axons had regrown in average four times longer in dKO compared with controls (Fig. 2c), as shown by whole-nerve Neurofilament immunofluorescence analyses. High magnifications delineated by six coloured boxes at comparable distances to the lesion site in control and dKO nerves showed in control nerves a Neurofilament signal of low intensity or fragmented starting from the second magnification closest to the lesion site (red box), whereas Neurofilament signal in dKO nerves remained mostly

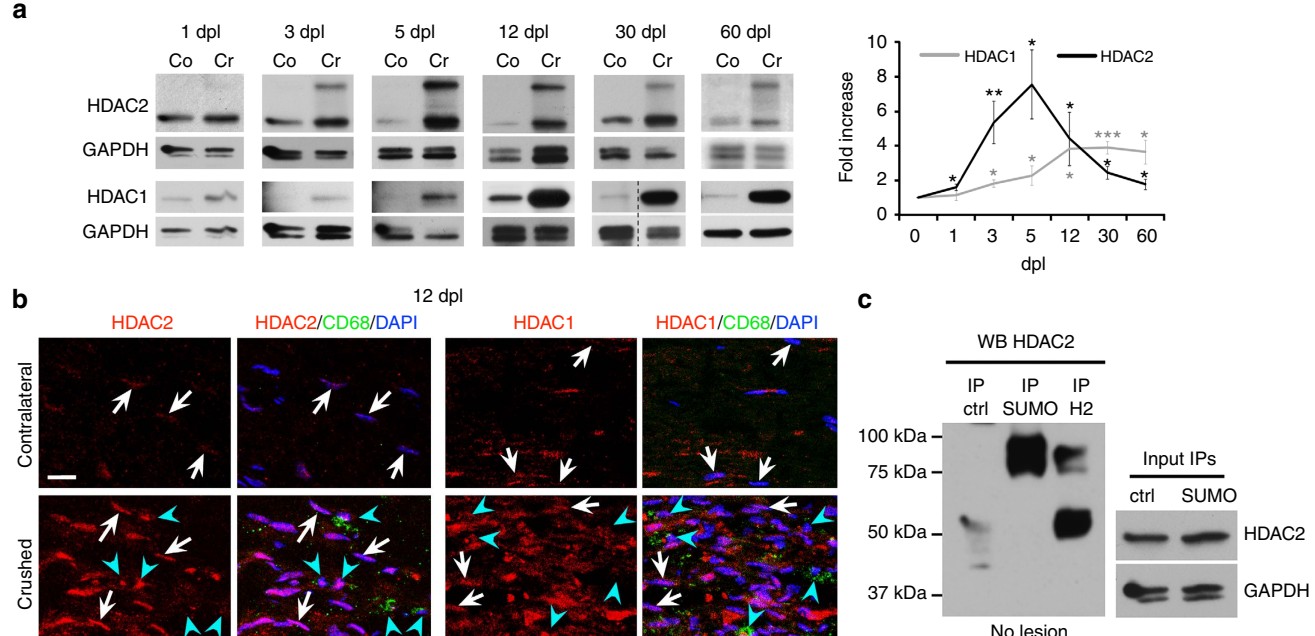

**Figure 1 | HDAC1/2 are robustly upregulated in SCs after lesion.** (**a**) Western blots of HDAC2 and HDAC1 in lysates of crushed (Cr) and contralateral (Co) nerves of adult mice at 1 dpl ($n = 5$), 3 dpl ($n = 5$), 5 dpl (HDAC2, $n = 6$; HDAC1, $n = 7$), 12 dpl (HDAC2, $n = 6$; HDAC1, $n = 8$), 30 dpl (HDAC2, $n = 6$; HDAC1, $n = 8$) and 60 dpl ($n = 6$), and quantification normalized to GAPDH and compared to Co $= 1$, showing HDAC1/2 upregulation after lesion. Of note: both bands detected by the HDAC2 antibody were quantified and added together in the graph. (**b**) Co-immunofluorescence of HDAC2 or HDAC1 (red) with CD68 (green = macrophages) and DAPI labelling (blue = nuclei, overlay appears pink) at 12 dpl, showing upregulation in SCs (CD68-negative cells with elongated nuclei, white arrows) of crushed compared to contralateral nerves. Note that macrophages (CD68-positive cells, blue arrowheads) express variable levels of HDAC1 and HDAC2. Scale bar, 10 μm. (**c**) Denaturing IP of SUMO-1, HDAC2 (H2) or control Flag (ctrl) in unlesioned (no lesion) adult mouse sciatic nerve lysates, and western blot of HDAC2, showing that the ~75 kDa band detected by the HDAC2 antibody corresponds to a SUMOylated form of HDAC2. HDAC2 and GAPDH western blots on lysates used for IP show the inputs. Representative photos of three independent experiments are shown. One-tailed (HDAC2, 12 dpl and 60 dpl; HDAC1, 5 dpl) or two-tailed Student's $t$-tests, unpaired (HDAC2: 3 dpl) or paired, $P$ values: $*P < 0.05$, $**P < 0.01$, $***P < 0.001$. Values, mean; error bars, s.e.m.

uninterrupted and at high intensity all along the imaged axons (Fig. 2c). In addition, the marker of axonal regrowth growth-associated protein 43 (GAP-43) was strongly increased in dKO compared with control nerves at 3 and 5 dpl (Fig. 2d). Axonal outgrowth was also increased by three to four times at 5 dpl (Fig. 2e), as shown by an increased number of axonal sprouts detected by electron microscopy. We next analysed control and dKO nerves at 1 month post lesion (mpl) to identify potential remyelination defects due to HDAC1/2 loss. We found that remyelinating sheaths were thinner in dKO, as measured by a higher g ratio (myelin diameter/(axon + myelin) diameter) compared to controls (Fig. 2f). These data indicate that HDAC1/2 slow down demyelination and axonal regrowth after lesion, but promote remyelination.

**HDAC1/2 deletion delays Oct6 but enhances cJun upregulation.** To understand the molecular mechanisms responsible for HDAC1/2 functions in delaying demyelination and axonal regrowth after lesion, we first determined the precise timing of transcription factors regulation after lesion. We chose to analyse expression levels of the myelination inhibitors cJun, Sox2 and Pax3, because of their previously reported involvement in SC dedifferentiation process after lesion[2,4] and because of the high quality of available reagents to quantify their expression by western blot. We also analysed expression levels of the intermediate inducer of SC differentiation Oct6, because of its known upregulation in SCs following a PNS lesion[10–12]. Consistent with previous findings[10–11], Oct6 was upregulated

very early after lesion, already at 1 dpl, at the mRNA level in control nerves (Supplementary Fig. 3). Oct6 protein levels were also upregulated at 1 dpl (Fig. 3a), before myelination inhibitors including cJun, Pax3 and Sox2 (Fig. 3b–d). Two Oct6 isoforms (one at ~42 kDa and a doublet at ~50–52 kDa) were upregulated at 1 dpl and detected after transfection with an Oct6-expressing construct (Supplementary Fig. 4a). Mass spectrometry confirmed both Oct6 isoforms detected by the anti-Oct6 antibody (Supplementary Fig. 4b). The heavier isoform may result from post-translational modifications of Oct6. Alternatively, the lighter isoform may result from cleavage, but the peptide coverage analysed by mass spectrometry did not allow to identify a potential cleavage, the most N-terminal and C-terminal detected peptides being both present in the two isoforms. Interestingly, while the total protein levels of Oct6 continued to be upregulated after 1 dpl until at least 12 dpl due to upregulation of the light isoform, the heavier isoform disappeared at 3 dpl and reappeared starting from 5 dpl. It is possible that these two Oct6 isoforms have different functions, but more work is needed to clarify this point. Oct6, which is highly expressed during development and downregulated in adult SCs (heavier isoform; ref. 14; Supplementary Fig. 4c), allows timely expression of the myelination inducer Krox20 during myelination and remyelination[12,13,15]. However, Krox20 upregulation after lesion occurs at a later time point as compared to Oct6 (ref. 9), suggesting another potential function of Oct6 upregulation at 1 dpl.

In dKO crushed nerves, Oct6 upregulation was delayed and levels were decreased compared with control crushed nerves

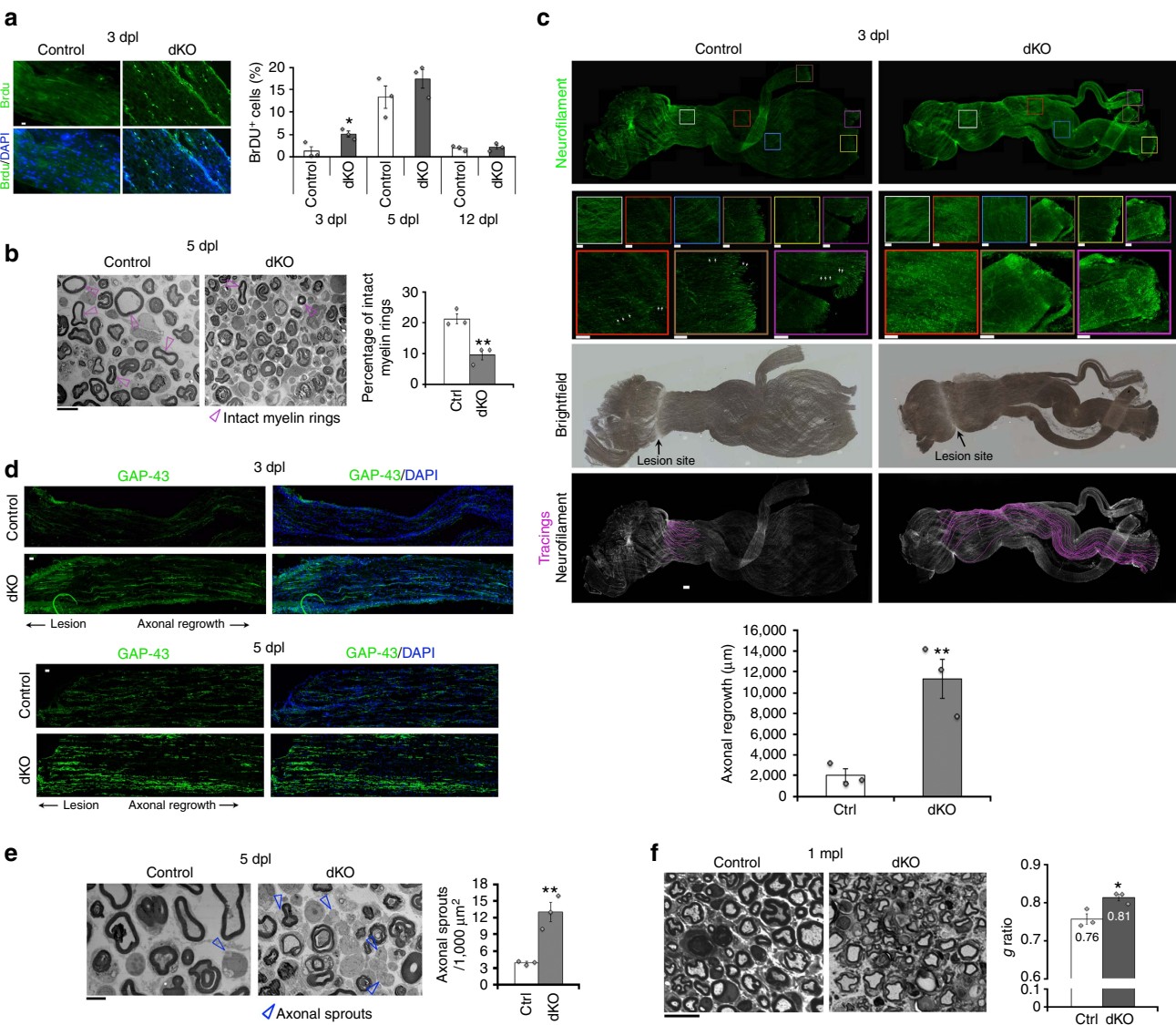

**Figure 2 | HDAC1/2 slow down axonal regrowth but promote remyelination. (a)** BrdU incorporation (green) and DAPI (blue) identifying earlier induction of proliferation in dKO compared to control nerves. Scale bar = 10 μm. Electron micrographs (**b,e**) and toluidine-blue staining (**f**) and quantification showing in dKO compared to controls a decreased percentage of intact myelin rings (**b**, scale bar = 10 μm) and increased number of axonal sprouts (**e**, Scale bar, 5 μm) at 5 dpl, and thinner remyelinating sheaths at 1 month post lesion (mpl) (higher g ratio in dKO, (**f**), Scale bar, 10 μm). (**c**) Z-series projections of whole-nerve Neurofilament immunofluorescence in control and dKO nerves, brightfield showing lesion site and axonal tracings (acquired using NeuronJ, Scale bar, 300 μm) quantifying axonal regrowth from the lesion site. White arrows, fragmented axons. Coloured boxes, magnifications (Scale bar, 100 μm). (**d**) Z-series projections of GAP-43 immunofluorescence (green) and DAPI labelling (blue = nuclei) showing strongly increased GAP-43 expression in dKO compared with control nerves at 3 dpl (20-μm thick cryosections) and 5 dpl (5-μm thick cryosections). Scale bar, 30 μm. Note that immunofluorescence staining of 3 dpl sections was not carried out at the same time as staining of 5 dpl sections, and imaging was also done separately with different exposure times. Unpaired two-tailed Student's t-tests, P values: *P < 0.05, **P < 0.01. Values, mean, error bars, s.e.m. Sample size: n = 3 animals per group per time point for all graphs, electron microscopy and immunofluorescence. (**a**) Number of cells counted: 334 to 469 per animal at 3 dpl (per genotype: 1,218 for controls, 1,140 for dKO); 111 to 727 per animal at 5 dpl (per genotype: 1,779 for controls, 1,262 for dKO); 553 to 4,380 per animal at 12 dpl (per genotype: 5,440 for controls, 8,892 for dKO). (**b**) Number of myelin rings counted: 344 to 469 per animal. (**c**) Number of tracings: 10 to 25 per nerve. (**d**) Surface of nerve ultrathin section counted: 0.026 to 0.03 mm$^2$ per animal. (**e**) 60–70 myelinated axons (randomly chosen) counted per nerve. The g ratio (axon diameter: axon + myelin diameter) of both contralateral (0.71 for control and 0.72 for dKO nerves, P value = 0.32: no significant difference between control and dKO) and crushed nerves was measured.

(Fig. 3a,e,f), while Oct6 levels were not affected in contralateral nerves of dKO compared with control mice, except for a mild decrease at 12 dpl (Supplementary Fig. 5a). Oct6 protein (Fig. 3a,e) and mRNA (Fig. 3f) were affected, indicating impairment at the transcriptional level. In contrast, cJun upregulation occurred earlier and to a higher level in dKO compared with control crushed nerves (Fig. 3b,g), while levels

were not affected in contralateral nerves of dKO compared with control mice (Supplementary Fig. 5b). Consistently, glial cell line-derived neurotrophic factor (GDNF), a cJun target gene, was upregulated earlier in dKO crushed nerves already at 3 dpl compared with control crushed nerves where expression was detectable later at 5 dpl (Supplementary Fig. 6). As for Pax3 and Sox2, expression levels remained mostly unaffected (Fig. 3c,d).

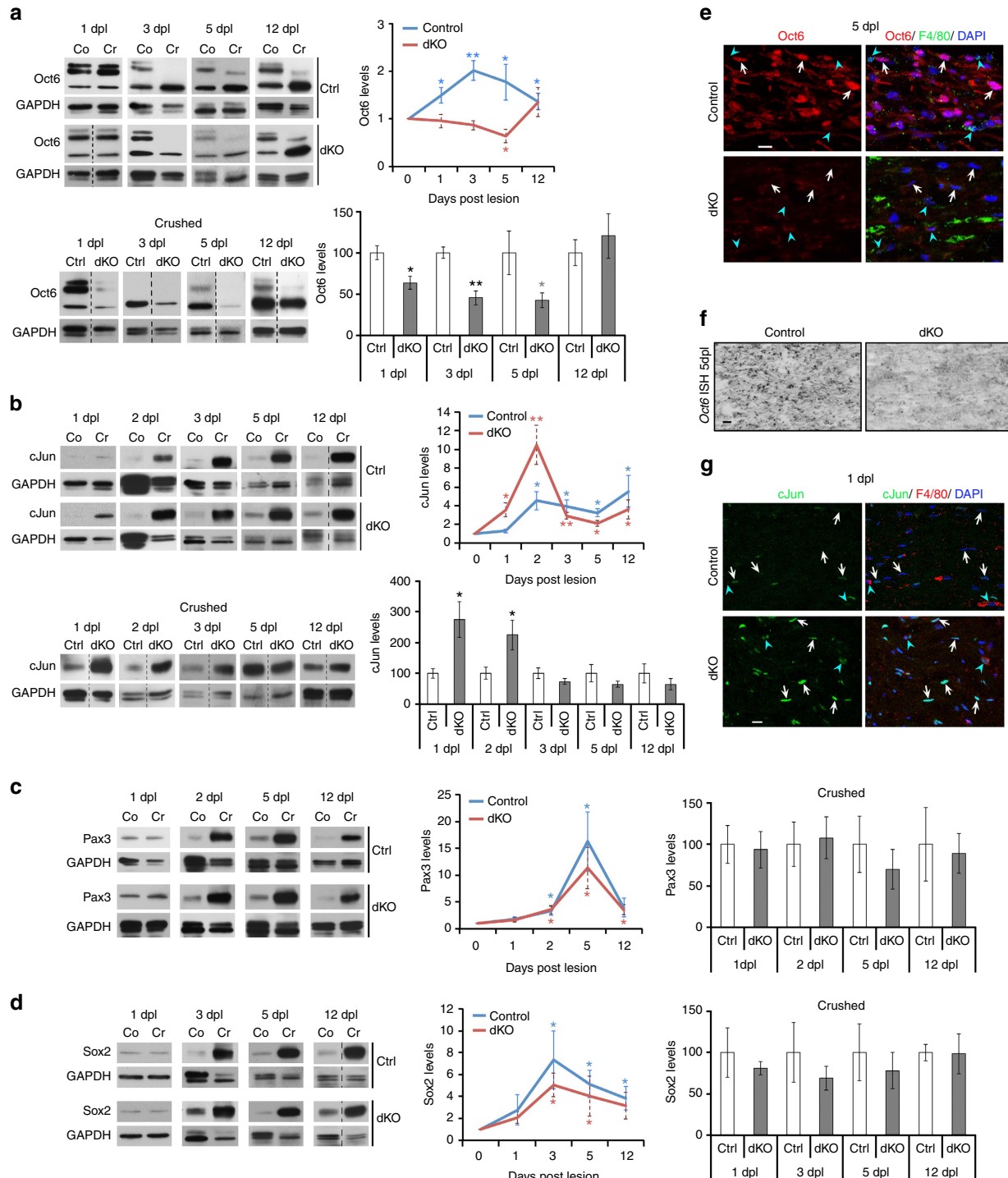

**Figure 3 | Delayed Oct6 and earlier/higher cJun upregulation in dKO.** Western blots and quantification normalized to GAPDH of (**a**) Oct6, (**b**) cJun, (**c**) Pax3, and (**d**) Sox2 in crushed (Cr) and contralateral (Co = 1) nerves of control (Ctrl) and dKO, showing delayed Oct6 upregulation and earlier/higher cJun upregulation in dKO. Of note: the double band at $\sim 52$ kDa and the lower band at $\sim 42$ kDa detected by the Oct6 antibody were quantified and added together in the graph. Dashed lines: samples run on the same gel, but not on consecutive lanes. In **a–d**, coloured graphs quantify protein levels in Cr compared to Co, and grey/white bar graphs quantify protein levels in dKO compared with Ctrl crushed nerves. (**e,g**) Co-immunofluorescence of Oct6 (red, **e**) at 5 dpl or of cJun (green, **g**) at 1 dpl with the macrophage marker F4/80 (green in **e**, red in **g**) and DAPI labelling (nuclei) in control and dKO nerves showing decreased Oct6 and increased cJun levels in SCs of dKO compared with control nerves. White arrows indicate SCs (F4/80-negative cells with elongated nuclei) and blue arrowheads indicate macrophages (F4/80-positive cells). (**f**) *Oct6 in situ* hybridization in control and dKO nerves at 5 dpl showing decreased Oct6 transcript levels in dKO. Sample size: (**a–d**) 5 animals per group for 1 dpl, 2 dpl and 3 dpl, and 6 or 9 animals per group for 5 dpl and 12 dpl; (**e–g**) 3 animals per group. One-tailed (grey asterisk; blue/red asterisks: Control Oct6 at 5 dpl, Control cJun at 5 dpl, dKO cJun at 12 dpl, Control Pax3 at 2 dpl, Control Sox2 at 3 dpl and 5 dpl) or two-tailed (black asterisks; blue/red asterisks: other values) Student's *t*-tests, unpaired (bar graphs, and cJun at 2 dpl in coloured graph) or paired (coloured graphs, except for cJun at 2 dpl), *P* values: *$P < 0.05$, **$P < 0.01$. Values, mean; error bars, s.e.m. Scale bar, 10 μm.

cJun is a major inducer of SC dedifferentiation and conversion into repair SCs, critically promotes axonal regrowth and survival, is required for functional recovery after PNS lesion, and is sufficient, when expression is enforced using adenoviral vectors, to restore regeneration kinetics after crush in nerves of Wld[s] slow regenerating mutant mice[7,8]. Thus, earlier SC conversion into repair cells and faster axonal regrowth in dKO is due at least partly to earlier and higher cJun upregulation compared with controls.

**Reduced Oct6 levels allow earlier conversion into repair SCs.** To analyse a potential causal relationship between delayed Oct6 and earlier/higher cJun upregulation, we used primary rat SCs (RSCs) and a dedifferentiation culture protocol (ref. 31; Supplementary Fig. 7). Oct6 downregulation by shRNA during RSC dedifferentiation increased cJun upregulation (Fig. 4a) and SC proliferation (Fig. 4b), but did not significantly affect Sox2 or Pax3 levels (Supplementary Fig. 8). Consistently, in Oct6 delta (Δ = deleted) SC enhancer (SCE) mice, where deletion of the SCE prevents Oct6 upregulation after lesion (ref. 12; Fig. 4c), cJun was upregulated earlier than in controls already at 1 dpl (Fig. 4d), and demyelination was increased at 5 dpl compared with controls (Fig. 4e, decreased percentage of intact myelin rings in Oct6 ΔSCE nerves), such as in dKO (Figs 3b and 2b, respectively). Also, consistent with dKO, axons had regrown significantly longer at 3 dpl (Fig. 4f). These data indicate that earlier cJun upregulation and conversion into repair SCs, and faster axonal regrowth in dKO mice after lesion are due at least partly to reduced Oct6 levels.

**HDAC1/2 deletion leads to reduced Krox20 and P0 expression.** We next analysed expression levels of promyelinating factors and myelin proteins at the redifferentiation and remyelination stages. At 12 dpl, Krox20 was upregulated (Fig. 5a) and remyelination started (data not shown). We found that protein (Fig. 5b,c) and mRNA (Fig. 5d) levels of Krox20 at 12 dpl were strongly reduced in dKO compared with controls, while expression levels of Sox10, the other major inducer of SC myelination, were not affected by the loss of HDAC1/2 (Fig. 5b). Similar to Krox20, expression levels of P0 mRNA (Fig. 5e) and protein (Fig. 5f) were strongly reduced at 1 mpl in dKO compared with control nerves. Krox20 is required for myelin formation and maintenance and for postnatal expression of P0 (refs 32,33), and P0 is an essential component of PNS myelin[34]. The thinner remyelination in dKO compared with control nerves is thus due at least partly to the failure of dKO SCs to upregulate Krox20 and P0 during redifferentiation and remyelination.

**HDAC2 recruits JMJD2C and KDM3A to Oct6 and Krox20.** We next aimed at elucidating HDAC1/2-dependent mechanisms controlling Oct6 upregulation after lesion. The Oct6 SCE contains two functionally interdependent conserved modules HR1 and HR2 sufficient to activate Oct6 developmental expression[35]. Exogenous DNA transiently applied to cells is present in nucleosomal DNA at about the same levels as endogenous DNA[36]. Therefore, we carried out luciferase gene reporter assays in RSCs to analyse whether HDAC1/2 modulate the Oct6 SCE activity during dedifferentiation. HDAC1 or HDAC2 overexpression increased the activity of the full Oct6 SCE and the ΔHR1 SCE, but not of the ΔHR2 SCE (Fig. 6a), suggesting that HDAC1/2 activate the Oct6 SCE HR2. By chromatin immunoprecipitations (ChIP), we found increased HDAC2 enrichment at the HR2 at 1 dpl compared with unlesioned (no lesion) nerves (Fig. 6b), while HDAC1 was not bound (Supplementary Fig. 9). Consistently, HDAC2 overexpression in

primary RSCs cultured under dedifferentiating conditions increased Oct6 mRNA levels (Fig. 6c).

HDACs have been described to act within multiprotein complexes that often contain HMTs or HDMs[26]. To understand the mechanism by which HDAC2 activates the Oct6 SCE HR2, we tested the potential effects of de-repressive K HDMs on Oct6 upregulation after lesion. Repressive K histone methylation marks are located on H3K9, H3K27 and H4K20. Among all known de-repressive K HDMs (data not shown), JMJD2C (H3K9me3 HDM) or KDM3A (H3K9me2/1 HDM) overexpression increased Oct6 mRNA levels in RSCs (Fig. 6c) and the activation of the full Oct6 SCE and the ΔHR1 SCE, but not of the ΔHR2 SCE (Fig. 6d), similar to the effect of HDAC2 overexpression. HDAC1/2 inhibition by Mocetinostat prevented JMJD2C- and KDM3A-mediated Oct6 SCE activation (Fig. 6d), indicating that this activation is HDAC1/2-dependent. To test the relevance of these findings in vivo, we first analysed the endogenous expression of JMJD2C and KDM3A in adult nerves after lesion and in primary RSCs cultured under dedifferentiation conditions. Compatible with potential functions in regulating Oct6 expression after lesion, JMJD2C and KDM3A were endogenously expressed in SC nuclear and cytoplasmic compartments after nerve lesion (Supplementary Fig. 10a,b) and also in dedifferentiated RSCs in culture (Supplementary Fig. 11a,b). Interestingly, they were both upregulated at 5 dpl (Supplementary Fig. 10a,b). By ChIP, we show in control nerves that JMJD2C is recruited to the HR2 at 1 dpl (Fig. 6e; Supplementary Fig. 12a) and KDM3A enriched at the HR2 in both unlesioned nerves and at 1 dpl (Fig. 6e; Supplementary Fig. 12b). In contrast, neither JMJD2C nor KDM3A were bound to the HR2 in dKO nerves (Fig. 6e). Consistently, H3K9me3/2 repressive marks were reduced in this region in control nerves at 1 dpl compared with unlesioned nerves (Fig. 6f), whereas in dKO H3K9me3 marks were increased (Fig. 6g). Interestingly, H3K9me2 marks were strongly reduced in dKO nerves (Fig. 6g), possibly due to some extent to impaired H3K9me3 demethylation. Taken together, these data demonstrate that HDAC2 allows JMJD2C and KDM3A recruitment to the Oct6 SCE HR2 for de-repression by H3K9 demethylation early after lesion.

We found strongly reduced Krox20 levels in dKO nerves at 12 dpl (Fig. 5b–d), at the time point where Krox20 is upregulated in control nerves to induce remyelination (Fig. 5a). Reduced Krox20 expression in the absence of HDAC1/2 may be partly due to delayed Oct6 upregulation after lesion. However, Oct6 levels are not significantly reduced at 12 dpl in dKO compared with control nerves. We thus asked whether HDAC1/2 directly regulate Krox20 expression after lesion. Indeed, at 12 dpl HDAC2, JMJD2C and KDM3A were recruited to the Krox20 MSE (myelinating Schwann cell element[15]; Fig. 6h), an essential enhancer for Krox20 expression in postnatal SCs. Consistently, H3K9me3/2 repressive methylation marks in this region were decreased at 12 dpl (Fig. 6h). Similar to their effect on the Oct6 SCE HR2, KDM3A or JMJD2C overexpression activated the Krox20 MSE in an HDAC1/2-dependent manner in RSCs cultured under redifferentiating conditions (Fig. 6i).

These data suggest that HDAC2, JMJD2C and KDM3A subsequently de-repress the Oct6 SCE HR2 and the Krox20 MSE after lesion to allow upregulation of Oct6 and Krox20.

**Oct6 and Krox20 activation by Sox10 depends on HDAC1/2.** The Oct6 SCE and Krox20 MSE are known targets of Sox10, a major transcription factor of SC differentiation[37], and we previously showed that Sox10 requires HDAC1/2 to upregulate Pax3 and P0 during SC specification[17], Sox10 itself and Krox20

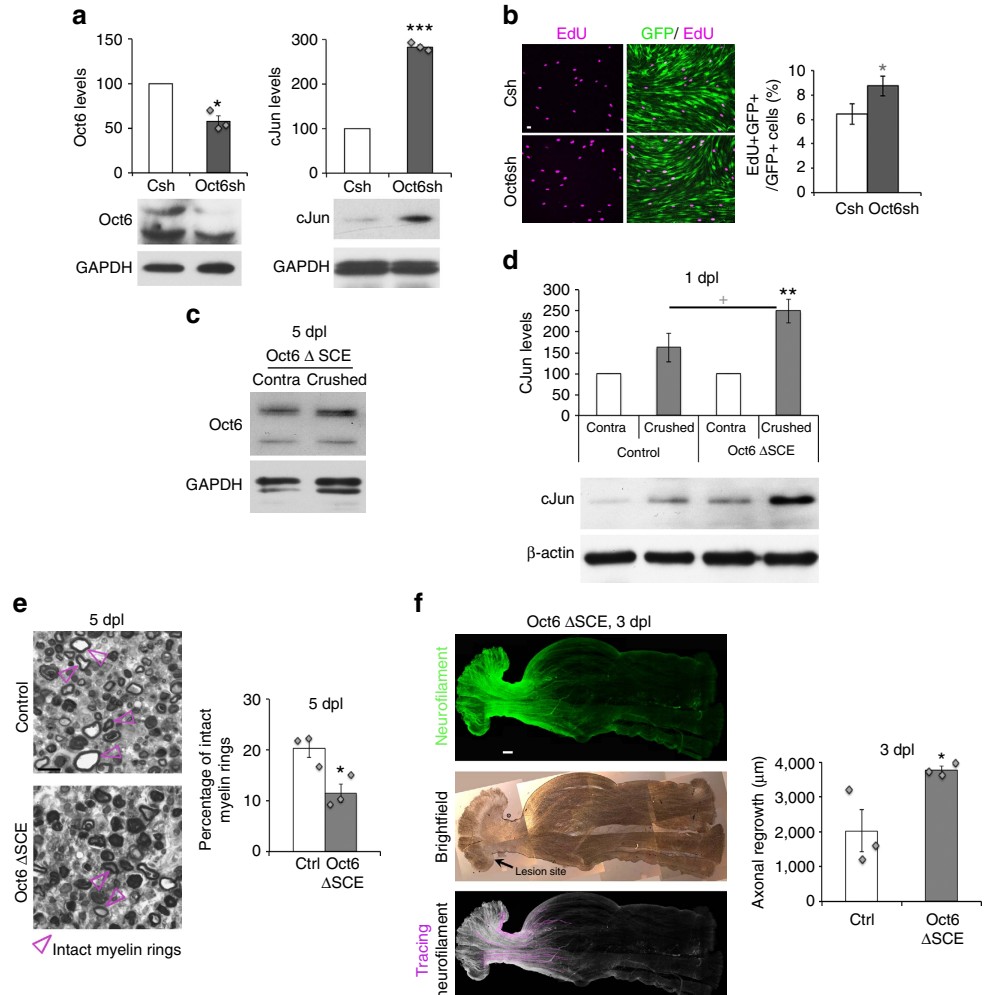

**Figure 4 | Decreased Oct6 expression accelerates axonal regrowth.** (**a**)Western blots of Oct6 and cJun in RSCs transduced with lentiviruses carrying control (Csh = 100) or Oct6 (Oct6sh) shRNAs and quantification normalized to GAPDH showing increased cJun levels in Oct6-downregulated RSCs. Western blot of (**c**) Oct6 in contralateral (contra) and crushed nerves of *Oct6* ΔSCE mice at 5 dpl (representative photos are shown) confirming absence of Oct6 upregulation in this mutant after lesion (as previously reported[12]), and of (**d**) cJun in crushed and contralateral nerves of adult control and *Oct6* ΔSCE mice at 1 dpl and quantification of protein levels in crushed nerves normalized to β-actin or GAPDH and compared to contralateral nerves ( = 100). (**b**) EdU incorporation identifying increased proliferation in RSCs transduced with lentiviruses carrying Oct6sh compared with Csh. Oct6sh and Csh lentiviruses also carry a GFP reporter showing high transduction efficiency in RSCs. Scale bar, 10 μm. (**e**) Toluidine-blue staining of semithin sections of *Oct6* ΔSCE and control mouse sciatic nerves at 5 dpl and quantification of intact myelin rings (pink arrowheads). Scale bar, 10 μm. (**f**) Z-series projections (confocal stacks) of whole-nerve Neurofilament immunofluorescence in *Oct6* ΔSCE nerves, brightfield showing lesion site and Neurofilament tracings (Scale bar, 300 μm) used for quantification of axonal regrowth (for this graph, the ctrl represented is the same as for graph in Fig. 2c). In *Oct6* ΔSCE nerves, cJun upregulation already occurred at 1 dpl, demyelination at 5 dpl was increased and axonal regrowth at 3 dpl was longer compared with controls, but these effects were less pronounced than in dKO nerves, possibly due to somewhat higher residual levels of Oct6 in the nerves of *Oct6* ΔSCE hypomorph mutant mice compared with dKO nerves. Two-tailed (black asterisks) or one-tailed (grey asterisk and cross) paired (**a**;**d**: crushed versus contra) or unpaired (**b**,**e**,**f**;**d**: *Oct6* ΔSCE crushed versus control crushed) Student's *t*-tests, P values: *P < 0.05 or $^{+}$P < 0.05; **P < 0.01, ***P < 0.001. Values, mean; error bars, s.e.m. In **d**, asterisk shows significance compared with contra and the cross compared with control. Sample size: (**a**) n = 3, (**b**) n = 9, (**c**) 3 animals per group, (**d**) 6 animals per group, (**e**) 3 animals per group, 227 to 435 myelin rings counted per animal. (**f**) 3 animals per group, 10 to 25 tracings per nerve.

during postnatal development[18]. We thus hypothesized that Sox10 coordinates the activation of the *Oct6* SCE and *Krox20* MSE after lesion with HDAC2-, JMJD2C- and KDM3A-mediated de-repression of these targets. Indeed, Sox10 recruitment to the *Oct6* SCE HR2 and *Krox20* MSE was increased at 1 and 12 dpl, respectively, compared with unlesioned nerves (Fig. 7a,b). In RSCs cultured under dedifferentiating conditions, Sox10 overexpression increased the ΔHR1, but decreased the ΔHR2 *Oct6* SCE activation (Fig. 7c), suggesting HR1 repression and HR2 activation by Sox10 during dedifferentiation. Sox10 also increased *Krox20* MSE activation in redifferentiating conditions (Fig. 7d). Sox10-induced activation of the HR2 (ΔHR1 construct) and the *Krox20* MSE was respectively fully and partly dependent on HDAC1/2 activity (Fig. 7c,d).

**Protein complex activating Sox10 target genes after lesion.** Consistent with HDAC1/2 known functions on histones, we found increased levels of acetylated histone H3 in dKO nerves at the *Oct6* SCE HR2 at 1 dpl and at the *Krox20* MSE at 12 dpl (Supplementary Fig. 13a,b), despite impaired *Oct6* and *Krox20* transcription. This indicates that histone acetylation or hyperacetylation in gene enhancers does not always correlate with transcriptional activation and supports the hypothesis that

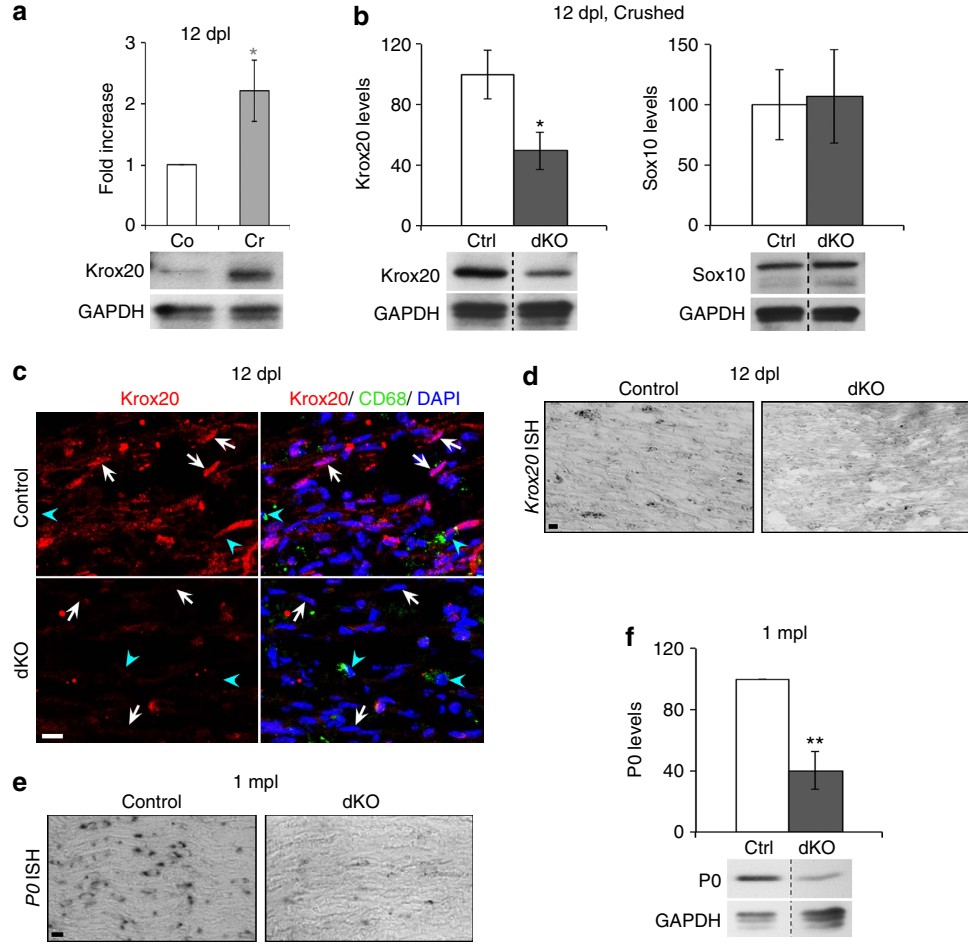

**Figure 5 | Reduced Krox20 and P0 expressions in dKO. (a)** Western blot of Krox20 in crushed (Cr) and contralateral (Co) nerves of control mice at 12 dpl and quantification normalized to GAPDH and compared with Co = 1. **(b,f)** Western blots of Krox20 and Sox10 at 12 dpl **(b)** and of P0 at 1 mpl **(f)** showing reduced Krox20 (but unaffected Sox10) and P0 levels in dKO compared with Ctrl crushed nerves at 12 dpl and 1 mpl, respectively. Dashed lines—samples run on the same gel, but not on consecutive lanes. **(c)** Co-immunofluorescence of Krox20 (red) with CD68 (green, macrophages) and DAPI labelling (blue, nuclei) in cryosections of control and dKO sciatic nerves at 12 dpl. White arrows indicate SCs (CD68-negative cells with elongated nuclei, blue arrowheads indicate macrophages (CD68-positive cells). *Krox20* **(d)** and *P0* **(e)** *in situ* hybridizations in control and dKO nerves showing decreased transcript levels in dKO nerves. Scale bar, 10 μm. One-tailed (grey asterisk) or two-tailed (black asterisks) Student's *t*-tests, unpaired **(b)** or paired **(a,f)**, *P* values: *$P < 0.05$, **$P < 0.01$. Values, mean; error bars, s.e.m. Sample size: **(a,b,f)** 6 animals per group, **(c–e)** 3 animals per group.

this may hamper the activation of some genes[21–23] or their de-repression. HDAC2 SUMOylation at lysine 462 has been shown to be required for HDAC2 activity and protein–protein interaction[38,39]. We thus generated an HDAC2K462R mutant and tested whether this mutation affects HDAC2 ability to activate the *Oct6* SCE and *Krox20* MSE. Indeed, HDAC2K462R overexpression strongly reduced the *Oct6* SCE and *Krox20* MSE activation (Supplementary Fig. 14), indicating that overexpressed HDAC2K462R overrode endogenous HDAC2 effects and that SUMOylation at lysine 462 is critical for HDAC2-dependent activation of these targets. We hypothesized that HDAC2, JMJD2C, KDM3A and Sox10 form a multifunctional protein complex that de-represses Sox10 targets and subsequently activates them. Indeed, SUMOylated HDAC2, JMJD2C, KDM3A and Sox10 interacted in unlesioned adult nerves (Fig. 8a–c), possibly to keep active Sox10-target genes such as *P0* for the maintenance of PNS integrity[20]. This complex was also assembled at 1 dpl and 12 dpl (Fig. 8a–c). Taken together, our data indicate that this complex is sequentially recruited to the *Oct6* SCE HR2 at 1 dpl and to the *Krox20* MSE at 12 dpl to de-repress and reactivate the transcription of these genes after lesion.

**HDAC1/2 inhibitor treatment accelerates functional recovery.** Faster axonal regrowth in dKO compared with controls supported a potential benefit of inhibiting HDAC1/2 during the first regeneration step after lesion. We thus treated mice with the HDAC1/2 inhibitor Mocetinostat for 3 or 5 days or with vehicle after lesion, to test whether this treatment accelerates regeneration kinetics. Mocetinostat is currently in Phase 2 clinical trials for the treatment of various cancers and did not induce detectable adverse effects during or after the treatments we carried out in mice. Mass spectrometry analyses on the nerves of mice treated with Mocetinostat revealed increased abundance of histone peptides compared with vehicle, after immunoprecipitation with an anti-acetyl-lysine antibody (Supplementary Fig. 15), suggesting efficiency of HDAC1/2 inhibition. Of note, HDAC1/2 are ubiquitously expressed, thus HDAC1/2 inhibitor treatment can potentially impact any cell type. A large majority (~85%) of nuclei present in the sciatic nerve at 1 dpl belongs to the Schwann cell lineage, while the remaining fraction can be inferred to endoneurial fibroblasts (~10%) and a few macrophages. Therefore, increased abundance of histone peptides and peptides of other nuclear proteins is likely due in large majority to HDAC1/2 inhibition in SCs; however, the effect of HDAC1/2

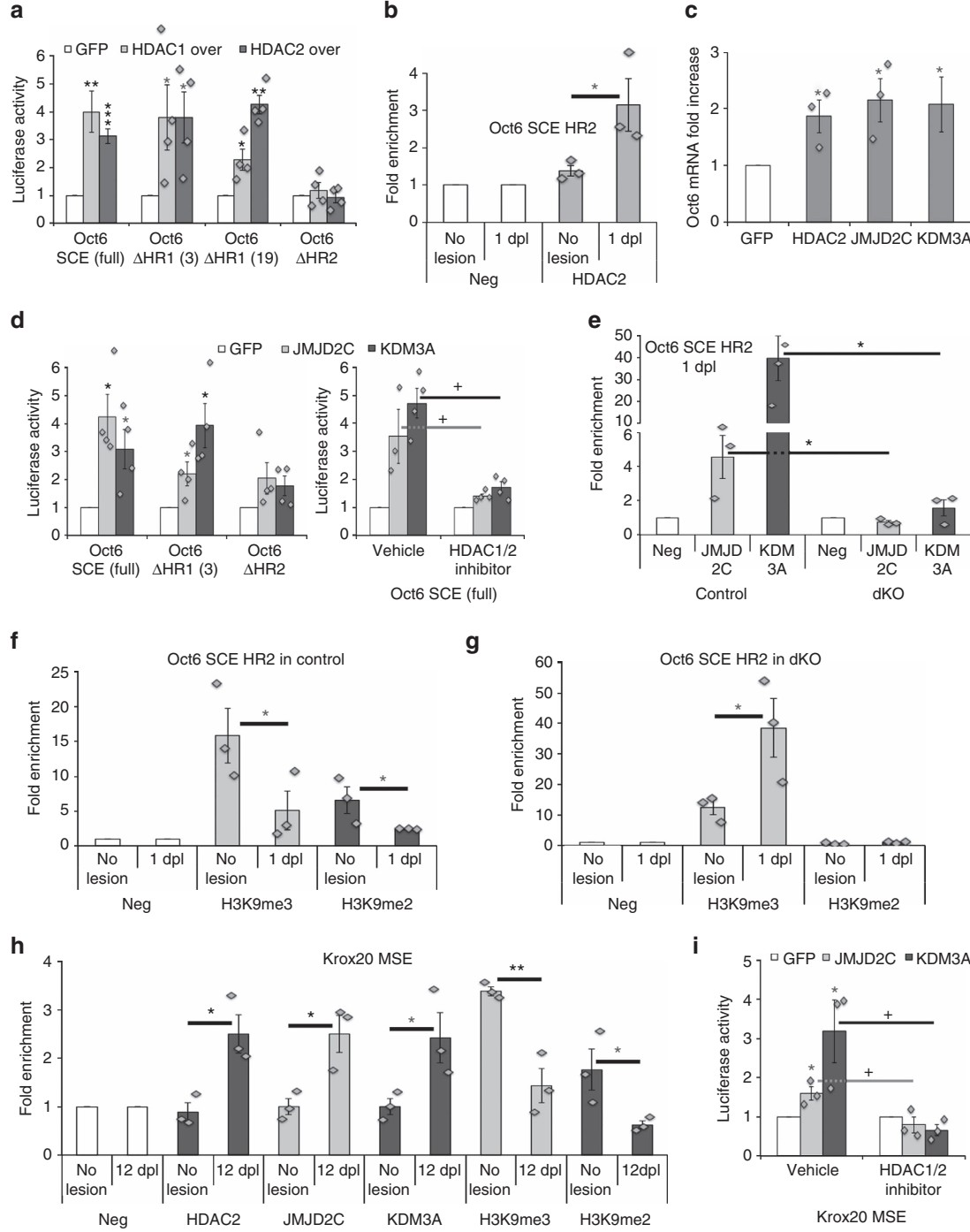

**Figure 6 | HDAC2-dependent de-repression by JMJD2C and KDM3A. (a,d)** Relative luciferase activity of the full *Oct6* SCE, the ΔHR1 (constructs #3 and #19, ref. 35) and the ΔHR2 SCE showing that overexpression of HDAC1, HDAC2, JMJD2C or KDM3A increases the activation of the full *Oct6* SCE or the ΔHR1 SCE, but not of the ΔHR2 SCE, compared with control GFP ( = 1) in RSCs cultured under dedifferentiating conditions. The experiment carried out in the presence of the HDAC1/2 inhibitor Mocetinostat or its vehicle indicates that activation induced by JMJD2C or KDM3A is prevented by HDAC1/2 inhibition. **(i)** Similarly, activation of the *Krox20* MSE induced by JMJD2C or KDM3A overexpression in RSCs cultured under redifferentiating conditions is prevented by HDAC1/2 inhibition. Chromatin immunoprecipitation of **(b)** HDAC2 and Flag (Neg = 1) in unlesioned (no lesion) nerves or at 1 dpl, of **(e)** JMJD2C, KDM3A and Neg (GFP or flag) in control or dKO nerves at 1 dpl, of **(f,g)** H3K9me3, H3K9me2 and Neg (GFP or Flag) in unlesioned nerves and at 1 dpl in control **(f)** and dKO **(g)** on the *Oct6* SCE HR2, and of **(h)** HDAC2, JMJD2C, KDM3A, H3K9me3, H3K9me2 and Neg on the *Krox20* MSE in unlesioned nerves or at 12 dpl. **(c)** Oct6 mRNA fold increase in RSCs transfected with GFP ( = 1), HDAC2, JMJD2C or KDM3A expression constructs and cultured under dedifferentiating conditions. Asterisks show significance compared with no lesion (**b,f–h**), to control (**e**) or to GFP (**a,c,d,i**), and crosses compared with vehicle (**d,i**). One-tailed (grey asterisks) or two-tailed (black asterisks or crosses) Student's *t*-tests, unpaired (**d,i**: HDAC1/2 inhibitor versus Vehicle; **b,e–h**) or paired (**d,i**: JMJD2C or KDM3A versus GFP; **a,c**), *P* values: *$P < 0.05$, +$P < 0.05$, **$P < 0.01$, ***$P < 0.001$. Values, mean; error bars, s.e.m. Sample size: (**a**) full *Oct6* SCE, $n = 12$; ΔHR1 and ΔHR2 SCE, $n = 4$; (**b**) 3 animals per group no lesion and 1 dpl; (**c**) HDAC2, JMJD2C, $n = 3$; GFP, KDM3A, $n = 6$; (**d**) $n = 4$; (**e**) $n = 3$ animals per group control and dKO; (**f,g**) 3 animals per group no lesion and 1 dpl; (**h**) 3 animals per group no lesion and 12 dpl; (**i**) $n = 3$.

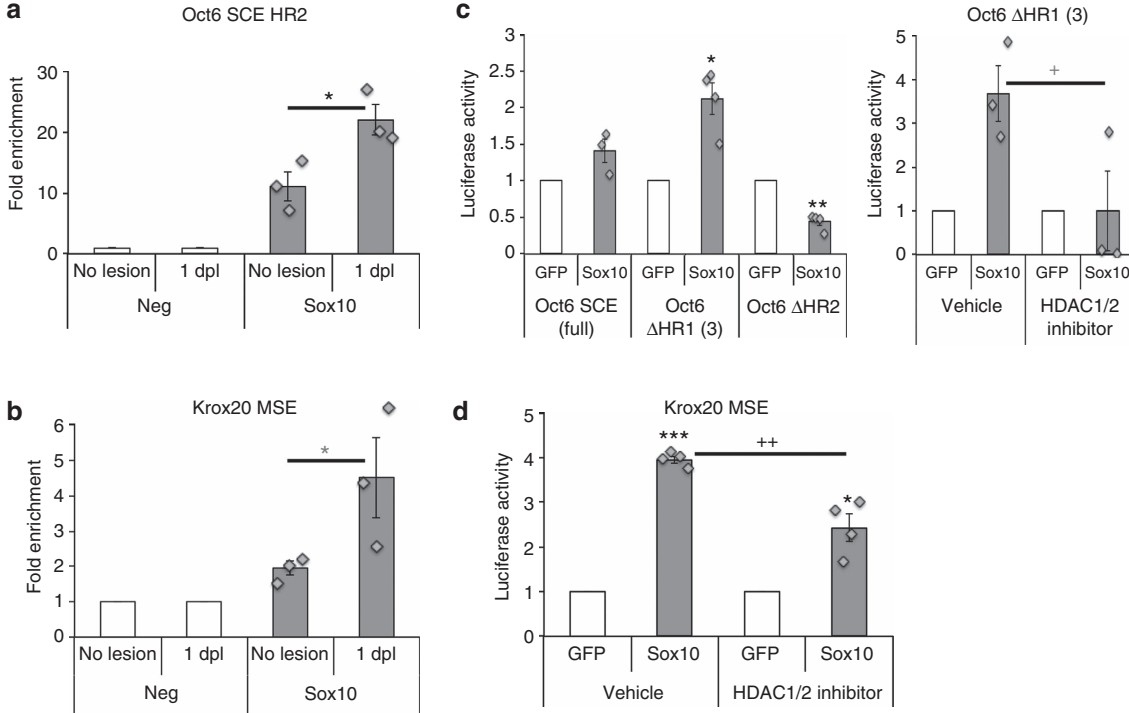

**Figure 7 | *Oct6* and *Krox20* activation by Sox10 depends on HDAC1/2.** Chromatin immunoprecipitation of Sox10 and GFP (Neg = 1) in (**a**) unlesioned (no lesion) nerves or at 1 dpl on the *Oct6* SCE HR2, or in (**b**) unlesioned nerves or at 12 dpl on the *Krox20* MSE. Relative luciferase activity of (**c**) the full Oct6 SCE, the ΔHR1 (construct #3; ref. 35) and ΔHR2 SCE, or of (**d**) the *Krox20* MSE, in primary RSCs transfected with GFP ( = 1) or Sox10 expression constructs and cultured under dedifferentiating (**c**) or redifferentiating (**d**) conditions for 1 day, in the presence of Mocetinostat (HDAC1/2 inhibitor) or its vehicle (**c,d**). One-tailed (grey asterisk or cross) or two-tailed (black asterisks or crosses) Student's *t*-tests, unpaired (**a,b;c,d**: HDAC1/2 inhibitor compared with Vehicle) or paired (**c,d**: Sox10 compared with GFP), *P* values: *$P < 0.05$, +$P < 0.05$; **$P < 0.01$; ++$P < 0.01$; and ***$P < 0.001$. Values, mean; error bars, s.e.m. Asterisks show significance compared with no lesion (**a,b**) or to GFP (**c,d**), and crosses compared with Vehicle (**c,d**). Sample size: (**a**) three animals per group no lesion and 1 dpl; (**b**) 3 animals per group no lesion and 12 dpl; (**c**) n = 3 or 4; (**d**) n = 4.

inhibition in fibroblasts and macrophages may contribute to some extent to the observed results. Three-day Mocetinostat-treated mice showed faster motor function and toe sensitivity recovery at 10 dpl (Fig. 9a,b), and at 18 dpl both Mocetinostat-treated groups had recovered toe sensitivity faster than vehicle-treated mice (Fig. 9b). Similar to dKO nerves, Oct6 was decreased and cJun increased at 1 dpl (Fig. 9c) and axons had regrown faster at 3 dpl (Supplementary Fig. 16) in nerves of Mocetinostat-treated mice compared with the vehicle group. Interestingly, this short-term Mocetinostat treatment resulted in improved regeneration at the morphological level: the density of sorted axons (mostly remyelinated) and of Remak bundle axons was increased in both Mocetinostat-treated groups compared to vehicle at 30 dpl (Fig. 9d). In addition, myelin thickness was subtly but significantly increased in nerves of 3-day Mocetinostat-treated mice as shown by a lower g ratio compared with the vehicle group (Fig. 9d), suggesting together with faster motor function recovery (Fig. 9a) that a 3-day treatment is optimal to improve overall regeneration efficiency.

## Discussion

After a PNS lesion, SCs not only dedifferentiate, but they also activate a repair programme that fosters axonal regeneration. The SC stage after lesion is thus distinct from the immature SC stage. This is why the concept of conversion into repair SCs or transdifferentiation has been recently established to provide a specific term corresponding to the cell cycle changes that SCs undergo after lesion. Repair SCs then redifferentiate to rebuild myelin of regenerated axons. These dynamic changes are associated with the regulation of several transcription factors that are thought or demonstrated to induce the different phases of the regeneration process in SCs, although the mechanisms controlling these regulations are not fully understood. In this study, we investigated the potential involvement of HDAC1 and HDAC2 in these mechanisms.

Our initial findings identifying a robust upregulation of HDAC1 and HDAC2 in SCs after lesion suggested important functions of these HDACs during the regeneration process. Indeed, ablation of HDAC1/2 in adult SCs led to impaired remyelination after lesion. This was consistent with previous studies of our group[18] and others[19] demonstrating critical functions of HDAC1/2 in developmental myelination of the PNS, by the control of Sox10, Krox20 and P0 transcription in postnatal SCs. In adult SCs after lesion, Sox10 levels were not affected by the ablation of HDAC1/2, but Krox20 re-expression was strongly reduced at the remyelination stage. In addition, we found that HDAC1/2—primarily HDAC2—also control the early phase of Oct6 upregulation after lesion, starting at 1 day post lesion. Considering the functions of HDAC2 and Oct6 in inducing SC differentiation, such an early upregulation of HDAC2 and Oct6 appeared counterintuitive. However, early upregulation of Oct6 in SCs after lesion confirmed previously reported observations[10,11]. Oct6 is critical for timely induction of Krox20 expression, but Krox20 is re-expressed to drive remyelination at around 12 dpl, at a much later stage as compared to Oct6 upregulation. This suggests that Oct6 is not only an inducer of Krox20 expression and may have other functions during regeneration. In support of this hypothesis, Oct6 is known to maintain SCs in a pre-myelinating stage[10,40],

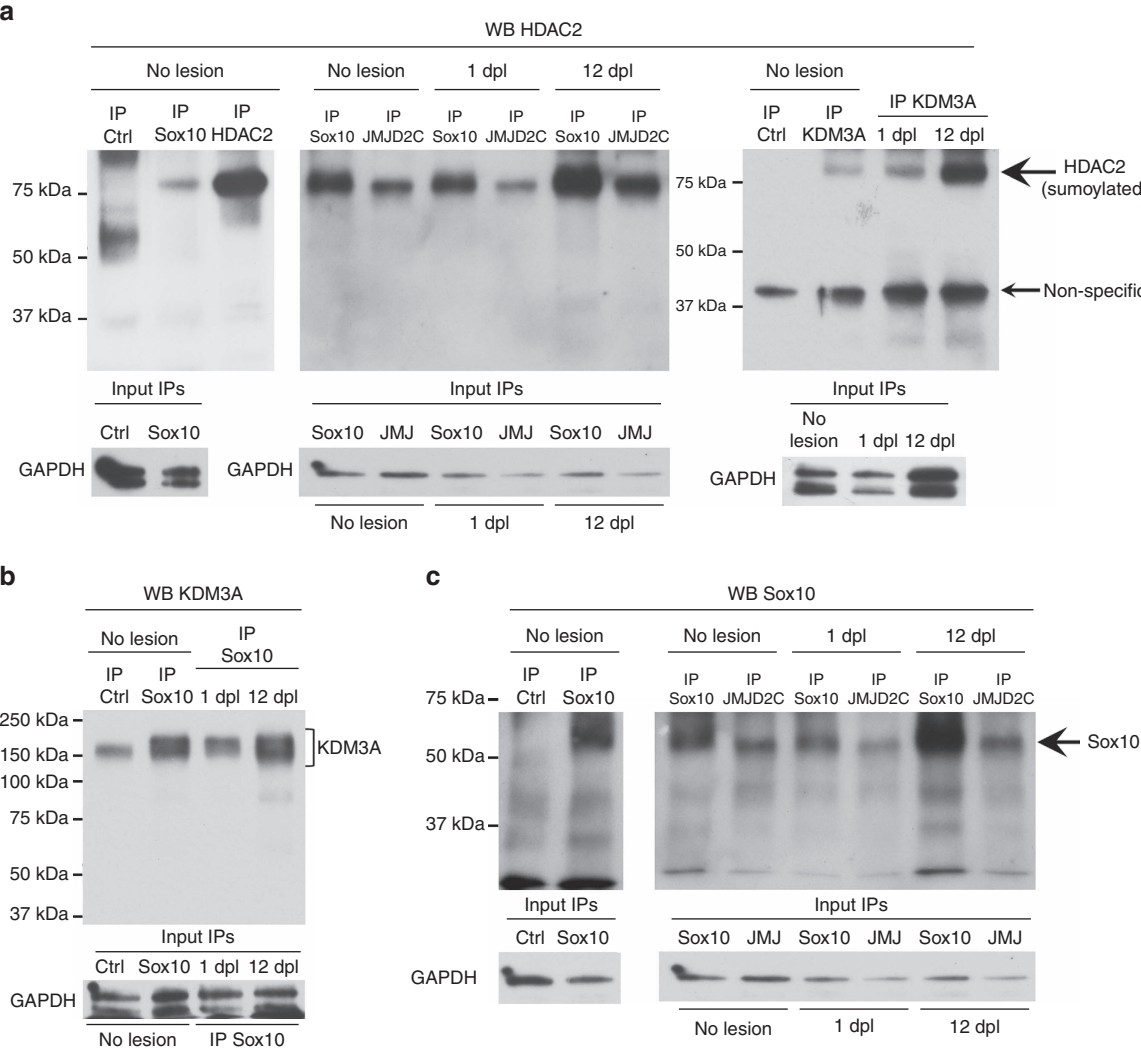

**Figure 8 | Assembly of multifunctional protein complex.** (**a–c**) Non-denaturing IP of Sox10, HDAC2, JMJD2C, KDM3A or ctrl (GFP or Flag) in unlesioned (no lesion) adult mouse sciatic nerve lysates or at 1 dpl or 12 dpl, and western blot of HDAC2 (**a**), KDM3A (**b**) or Sox10 (**c**). Membranes where Sox10 and JMJD2C IPs were run together were first blotted with the HDAC2 antibody (**a**) and were re-blotted with the Sox10 antibody (**c**). GAPDH western blots on lysates used for IP show the inputs (in **a**, only one input for no lesion IP KDM3A and ctrl that were done on the same lysate divided by two). Sample size: each IP was done three times, using nerves of three different animals. One nerve was used per IP.

its downregulation being necessary for myelination to proceed[16]. However, although the underlying molecular mechanisms remain to be elucidated, our findings indicate that early upregulation of Oct6 after lesion transiently counteracts the upregulation of cJun, a key inducer of SC dedifferentiation and conversion into repair cells. This may indicate that early Oct6 upregulation after lesion triggers the very start of the SC redifferentiation programme and act as a priming event for later remyelination. Alternatively, early Oct6 upregulation after lesion may be the result of an innate SC reaction to injury attempting to maintain the myelinating state. This would somewhat resemble the transient upregulation of myelin proteins observed in oligodendrocytes, the myelinating glia of the CNS, in response to a CNS injury[4,41]. In any case, we show here that early upregulation of Oct6 after lesion is not necessary and even acts as a brake for the regeneration process. Indeed, our study demonstrates that the innate SC response to injury upregulating HDAC2 early after lesion is not optimal for regeneration speed, this speed being substantially increased by short-term inhibition of HDAC1/2 after lesion. This prevents early Oct6 upregulation and results in earlier cJun upregulation and faster recovery of motor and sensory functions correlating with faster axonal outgrowth and increased density of regenerated axons. Interestingly, the shortest HDAC1/2 inhibitor treatment we carried out (3-day Mocetinostat) also subtly increased myelin thickness at 1 mpl, suggesting that faster axon regeneration may allow earlier remyelination. However, this is speculative and needs to be further tested. A longer Mocetinostat treatment of 5 days also led to increased density of regenerated axons at 1 mpl and to faster sensory function recovery, but less fast than the 3-day treatment, and did not lead to faster motor function recovery, nor to increased myelin thickness. We hypothesize that a longer treatment with the HDAC1/2 inhibitor may slow down remyelination, but this also needs to be further tested.

Besides the potential medical applications of our findings, we were very interested in understanding the molecular basis of HDAC1/2-mediated activation of Oct6 expression. Oct6 transcription in SCs depends on the activation of the SCE, a SC-specific enhancer located ∼10 kb downstream of Oct6 transcriptional start site. The SCE comprises two functionally interdependent conserved modules called HR1 and HR2 that contain Sox10 binding sites. We found that overexpression of HDAC1 or HDAC2 activates the Oct6 SCE HR2 in primary RSCs

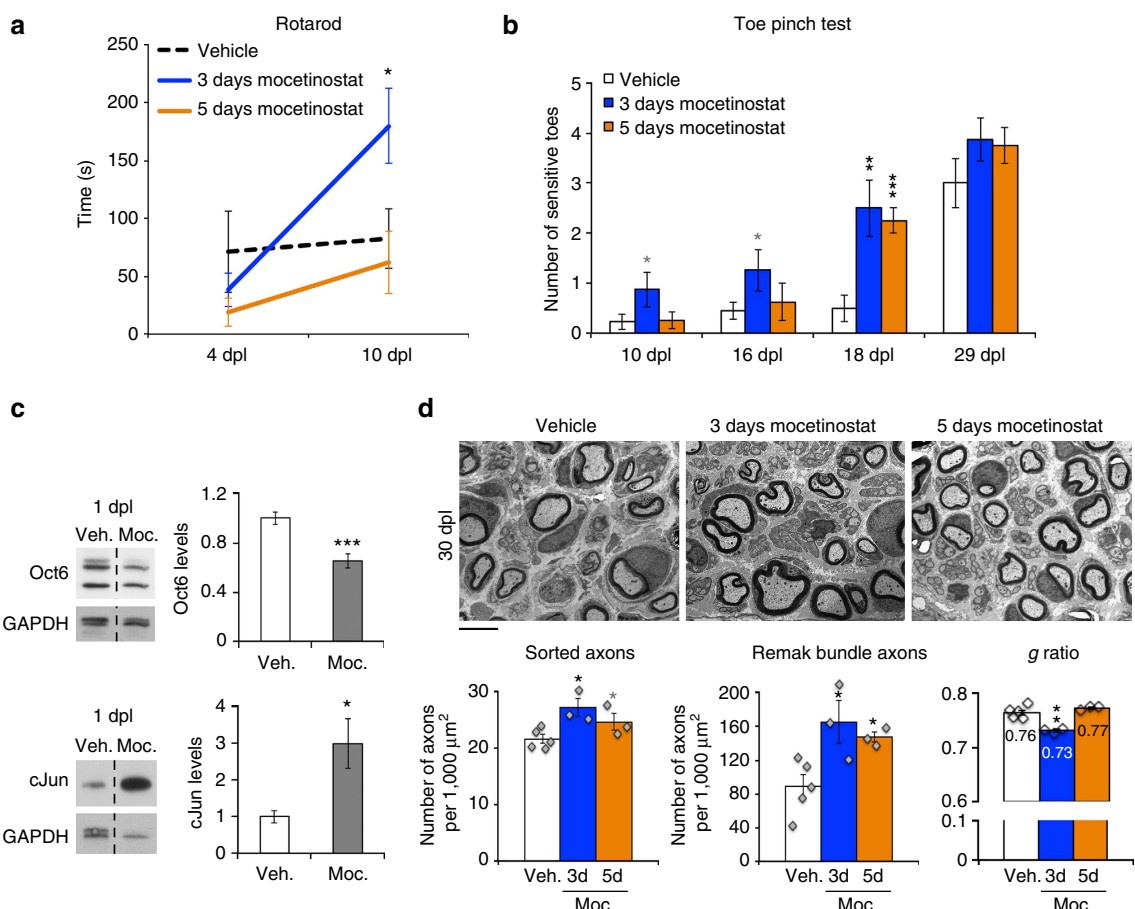

**Figure 9 | HDAC1/2 inhibitor treatment accelerates functional recovery.** Quantification of motor function recovery by Rotarod test at 4 and 10 dpl (**a**) and of sensory function recovery by toe pinch test at 10, 16, 18 and 29 dpl (**b**) in vehicle, 3-day or 5-day Mocetinostat-treated mice after sciatic nerve lesion. (**c**) Western blot of Oct6 and cJun and quantification normalized to GAPDH showing reduced Oct6 and increased cJun levels in nerves of mice treated with Mocetinostat (Moc.) for 1 day after lesion compared to vehicle (Veh.). (**d**) Electron micrographs of nerve ultrathin sections from vehicle, 3-day and 5-day Mocetinostat-treated mice and quantification showing increased density of sorted (mostly myelinated) axons and Remak bundle axons in both Mocetinostat-treated groups and increased myelin thickness at 30 dpl in the 3-day Mocetinostat-treated group, compared with vehicle (Veh.). Scale bar, 5 μm. One-tailed (grey asterisks) or two-tailed (black asterisks) unpaired Student's *t*-tests, *P* values: *$P < 0.05$, **$P < 0.01$, ***$P < 0.001$. Values, mean; error bars, s.e.m. Asterisks show significance compared with the vehicle group. Sample size: (**a,b**) Vehicle, 9 animals; 3-day Mocetinostat, 8 animals; 5-day Mocetinostat, 8 animals; (**c**) 6 animals per group; (**d**) Vehicle, 5 animals; 3-day Mocetinostat, 3 animals; 5-day Mocetinostat, 3 animals.

cultured under dedifferentiating conditions. However, only HDAC2 was recruited to the HR2 at 1 dpl in mouse nerves, correlating with its early upregulation after lesion, whereas HDAC1 is upregulated later. HDAC1/2 are known to belong to various chromatin-remodelling complexes that often include other chromatin-remodelling enzymes, such as HMTs or HDMs. Most of these complexes have been reported to silence rather than activate gene transcription, thus we set out to identify novel protein complexes that can activate transcription.

We found that the H3K9 HDMs JMJD2C and KDM3A are recruited/enriched at the *Oct6* SCE HR2 in an HDAC1/2-dependent manner at 1 dpl to demethylate this region of the SCE and allow its de-repression by H3K9 demethylation. The same HDMs were also recruited to the *Krox20* MSE at 12 dpl when Krox20 is upregulated to induce remyelination. The *Oct6* SCE and the *Krox20* MSE are direct targets of the transcription factor Sox10, and we previously found that HDAC2 interacts with Sox10 to activate different Sox10 target genes during SC development. Here we identify the assembly of a multifunctional protein complex containing HDAC2, JMJD2C, KDM3A and Sox10 in adult nerves, with the likely function of keeping active Sox10 target genes necessary for the maintenance of PNS

integrity, such as P0 (ref. 20). After lesion, this complex is subsequently recruited to the *Oct6* SCE HR2 and the *Krox20* MSE for their de-repression and activation.

Interestingly, we show that HDAC2 is SUMOylated in adult nerves and that SUMOylation is increased after lesion. In mammalian cells, HDAC2 is known to be SUMOylated by small ubiquitin-related modifier (SUMO)-1 (ref. 38). SUMOylation is achieved by an enzymatic machinery consisting of an E1-activating enzyme (SAE1 or SAE2), the SUMO E2-conjugase Ubc9 and a SUMO E3-ligase (15 mammalian enzymes currently known). This machinery catalyses the covalent attachment of SUMO to lysine residues that can be removed by SUMO-specific proteases (seven known enzymes SENP-1 to SENP-7). SUMOylation is known to modify the activity, binding partners and/or stability of proteins[27]. HDAC2 SUMOylation at K462 has been previously demonstrated to be necessary for HDAC2 activity and protein–protein interaction, in the context of HDAC2 binding and deacetylation of p53 (ref. 38). Consistent with this, we show here that SUMOylated HDAC2 is the detectable form of HDAC2 interacting with JMJD2C, KDM3A and Sox10 in adult sciatic nerves, and overexpression of the SUMOylation-deficient HDAC2 mutant HDAC2K462R decreases the activity of the *Oct6*

SCE HR2 and the *Krox20* MSE, thus acting as a dominant-negative HDAC2 mutant. These results thus identify a critical function of SUMOylation for HDAC2 activity on its targets after lesion.

What remains to be elucidated is the direct deacetylation target of HDAC1/2 that allows the recruitment of JMJD2C and KDM3A to the *Oct6* SCE HR2 and the *Krox20* MSE. As expected from the known function of HDAC1/2 on histones, we found hyperacetylation of histone H3 at the *Oct6* SCE HR2 and the *Krox20* MSE at 1 and 12 dpl, respectively, in dKO nerves, despite impaired transcription of *Oct6* and *Krox20* at these time points. These data indicate that hyperacetylation of histones in gene regulatory regions does not always correlate with increased transcriptional activation. This is consistent with previous studies showing that HATs and HDACs are simultaneously recruited to active genes to provide an appropriate level of histone acetylation for gene elongation, and to reset chromatin to maintain genes in an active state[21,22]. Further investigations are needed to identify whether histone deacetylation or deacetylation of another protein by HDAC1/2 critically regulates JMJD2C and KDM3A recruitment to target genes after lesion.

In summary, we demonstrate that, after a PNS lesion, SUMOylated HDAC2, JMJD2C and KDM3A collaborate in adult SCs to sequentially de-repress *Oct6* and *Krox20*, two Sox10 target genes critical for SC development and redifferentiation programme after lesion (mechanism summarized in Fig. 10). This mechanism acts as a brake for SC conversion into repair cells and axonal regrowth, but drives remyelination. This brake can be released by short-term treatment with an HDAC1/2 inhibitor after lesion, thereby accelerating regeneration of peripheral nerves and identifying a new therapeutic strategy for the improvement of peripheral nerve regeneration after lesion.

## Methods

**Statistical analyses.** For each data set presented, experiments were performed at least three times (exact n indicated in each figure legend) and $P$ values were calculated using two-tailed (black asterisks and crosses in the figures) or one-tailed (grey asterisks or crosses in the figures) Student's $t$-tests. $P$ values: $*P < 0.05$, $+P < 0.05$; $**P < 0.01$, $++P < 0.01$; and $***P < 0.001$, $+++P < 0.001$. For data sets obtained using animals or their tissues, three to nine animals were used per group and tissues of each animal were processed independently. For data sets obtained using cells, three to 12 independent experiments were conducted and analysed. For animal experiments, sample size was determined by the minimal number of animals required to obtain statistically significant results, and increased in some cases to improve confidence in the results obtained ($n = 3–5$ for microscopy, $n = 5–8$ for western blot analyses, $n = 8–9$ for behavioural analyses). No animal or sample was excluded from the analysis.

**Animals.** Tamoxifen-inducible Schwann cell-specific HDAC1/2 knockout mice (dKO[20]) and *Oct6* ΔSCE mice[12] have been previously described. To temporally induce gene ablation in dKO adult mice (∼30 g body weight), we administered 2 mg tamoxifen intraperitoneally (i.p.) daily for 5 days. As control mice, we used tamoxifen-injected littermate mice of the same sex whenever possible, and at least tamoxifen-injected age-matched mice of the same sex with similar genetic background (from breedings of the same mouse lines). Surgery: we used Isoflurane (3% for induction, 1.5–2% for narcosis during operation) for anaesthesia. For analgesia, 0.1 mg kg$^{-1}$ per body weight buprenorphine (Temgesic; Essex Chemie) was administered by i.p. injection 1 h before nerve lesion and after the operation every 12 h during 3 days. The field of operation was cleaned and disinfected. An incision was made at the height of the hip and the sciatic nerve was exposed on one side. The nerve was crushed ($5 \times 10$ s with crush forceps: Ref. FST 00632-11). The wound was closed using Histoacryl Tissue Glue (BBraun). After the operation, mice were wrapped in paper towels and placed on a warming pad until recovery from anaesthesia. When necessary, Mocetinostat (HDAC1/2 inhibitor) or its vehicle was injected in the pelvic cavity at 10 mg kg$^{-1}$ after wound closing and mice were sacrificed either 24 h later or treated again once a day for 2 days (3-day treated) or 4 days (5-day treated) for functional recovery experiments. In some cases, BrDU (100 mg kg$^{-1}$ body weight) was injected i.p. two hours before killing to mark proliferating cells. Before collection of sciatic nerves, mice were killed by a lethal i.p. injection of 150 mg kg$^{-1}$ Pentobarbital (Esconarkon; Streuli Pharma AG). Male and female mice of mixed strains and between 3 and 4 months of age were used. No randomization method was used, but experimenters were blinded to the experimental group (genotype, treatment) and received only the animal number given at birth by the animal caretaker. Animal use was approved by the Veterinary office of the Canton of Fribourg.

**Functional recovery experiments.** Mice were placed three times on the Rotarod apparatus at a fixed speed of 15 r.p.m. to test balance and motor coordination, at 4 and 10 days post lesion. The duration of each trial was limited to 300 s, and trials were separated by a 30 min recovery period. Latency to fall from the rotating beam was recorded and the average of the three trials was used for quantification. Recovery of sensory function was tested at 4, 10, 16, 18 and 29 dpl by toe pinch test: each toe of the rear foot on the right side (lesioned side) was pinched with equal pressure applied by the same experimenter using flat tip forceps. Immediate withdrawal was recorded as functional sensitivity of the pinched toe. In case no toe exhibited sensitivity, the same test was applied to toes of the contralateral side (uninjured side), which always resulted in immediate withdrawal. All tests were carried out with the same experimental animals (16 Mocetinostat-treated mice: 8 for the 3-day treatment group and 8 for the 5-day treatment group, and 9 vehicle-treated mice). All mice were between 3 and 4 months old. The experimenter was blinded regarding the treatment that mice received.

**Constructs.** Expression constructs: pBJ5 HDAC1 expression construct (kind gift from S. Schreiber), pME1 8 S HDAC2 expression construct (kind gift from E. Seto), Sox10 expression construct (kind gift from M. Wegner), p3xFlag-CMV-JMJD2C (kind gift from Adolfo Saiardi), pcDNA4-FLAG-Jhdm2a (ref. 42; = KDM3A, gift from Toshinobu Nakamura & Toru Nakano, Addgene plasmid #38136). HDAC2K462R was generated by site-directed mutagenesis, using pME1 8 S HDAC2 as template. Primers (5′-phosphorylated) were as follows: forward, 5′-AAGAAGACAGATGTTAGGGAAGAAGACAAATCC-3′, reverse, 5′-GGATT TGTCTTCTTCCCTAACATCTGTCTTCTTG-3′.

Luciferase constructs: *Oct6* full SCE, *Oct6* ΔHR1 SCE #3 and #19 (ref. 35; kind gifts from Dies Meijer), Krox20-MSE8 pGL3-TATA[43] (gift from Jerry Crabtree, Addgene #21261). We generated *Oct6* ΔHR2 SCE by PCR using the following primers: forward, 5′-CCGGCCGATATCTCAATGACTCACTCCTGATGG-3′, reverse, 5′-CCGGCCGTCGACTCCTTTCAGCCTTCTGTTCC-3′. The PCR product was digested with EcoRV and SalI, and inserted into pGL3-promoter vector (Promega) that was first digested with BamH1, blunted and digested again with SalI.

Constructs for *in situ* hybridization probes: pBluescript-Krox20 (kind gift from Piotr Topilko), P0 (kind gift from G. Lemke and R. Axel), pBluescript-Oct6 (kind gift from Dies Meijer). Newly generated constructs were verified by sequencing.

**Cell culture.** Purified primary rat Schwann cell (RSC) cultures were obtained as described[44]. Identity and purity was checked for each primary preparation by immunofluorescence of SC-specific markers (p75, Sox10, Oct6, Krox20, P0, MAG) under proliferation and differentiation conditions. Mycoplasma contamination was not tested, because of the low incidence of mycoplasma contamination in primary cells, and because mycoplasma contamination results in inefficiency of transfection, which we did not observe in our primary cells. RSCs were grown in proliferation medium: DMEM containing 10% FCS (Gibco), 1:500 penicillin/streptomycin (Invitrogen), 4 µg ml$^{-1}$ crude GGF (bovine pituitary extract, Bioconcept), and 2 µM forskolin (Sigma), at 37 °C and 5% CO$_2$/95% air. Schwann cell dedifferentiation and redifferentiation culture protocols were previously described[31]. Briefly, RSCs were first growth-arrested in defined medium (DM[44]) for 8 to 15 h, then 1 mM dbcAMP (Sigma) was added to induce differentiation. Cells were incubated in this medium for another 3 days. The medium was then changed to DM only without dbcAMP, and incubated in this medium for 3 days (differentiation mimicking adult Schwann cell stage). To induce dedifferentiation, cells were then changed to proliferation medium and incubated in this medium for 24 h (dedifferentiating conditions) or 3 days (dedifferentiation). For redifferentiating conditions, cells under dedifferentiation were changed to DM and incubated for 8 to 15 h, then 1 mM dbcAMP was added and cells were incubated for another 24 h in this medium.

**Luciferase gene reporter assay.** This assay was conducted as previously described[18]. When cells were assayed in dedifferentiating conditions, transfection was carried out at the time of change to proliferation medium. When cells were assayed in redifferentiating conditions, transfection was carried out at the time of dbcAMP addition after dedifferentiation. Cells were lysed 24 h after transfection. For some experiments, 0.6 µM Mocetinostat (at this concentration, Mocetinostat is a specific inhibitor of HDAC1/2: IC50 for HDAC1 = 0.15 µM, IC50 for HDAC2 = 0.29 µM; at higher concentrations, Mocetinostat is also active on HDAC3: IC50 for HDAC3 = 1.66 µM) or its vehicle was added to the cells at the time of transfection.

**qRT-PCR.** Isolation of RNA was carried out using Trizol reagent (Invitrogen) and cDNA was produced using M-MLV Reverse Transcriptase (Promega), according to the manufacturers' recommendations. Quantitative real-time PCR analyses were performed with an ABI 7000 Sequence Detection System (Applied Biosystems)

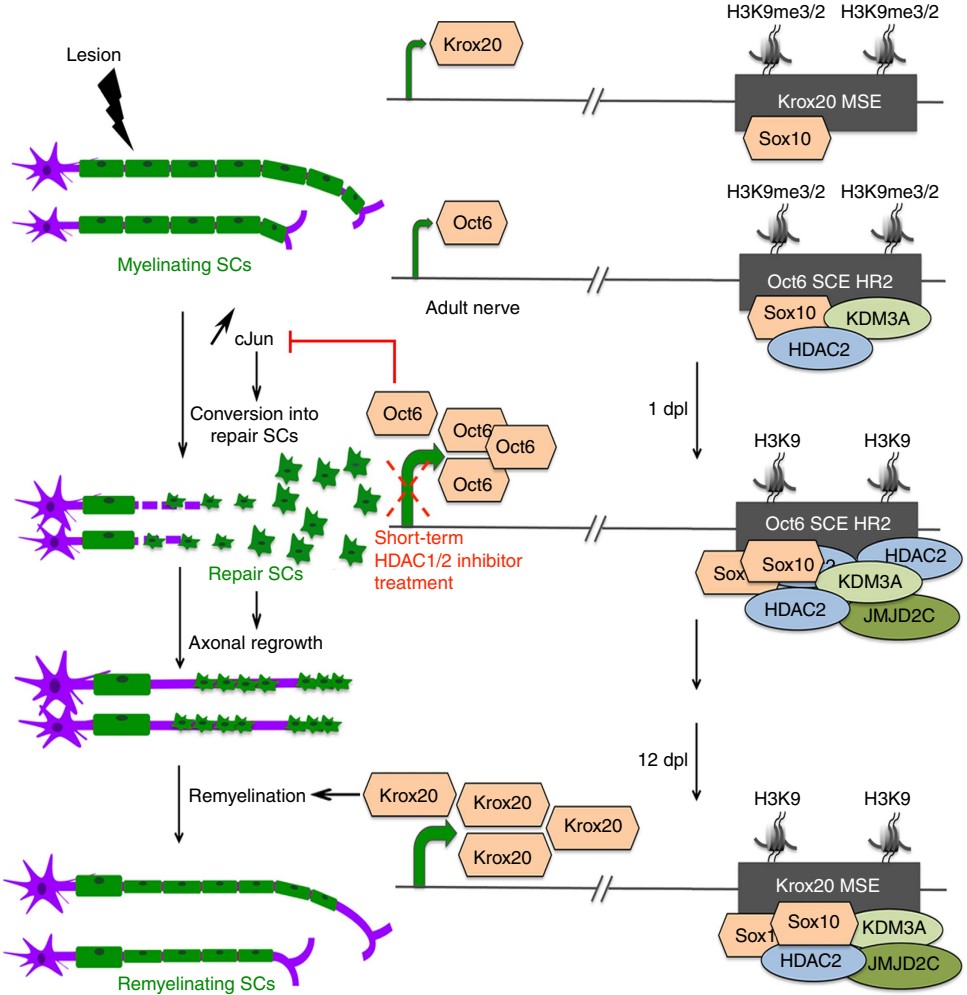

**Figure 10 | Mechanism of *Oct6* and *Krox20* de-repression after lesion.** In myelinating SCs of adult peripheral nerves, low levels of Sox10 are bound to the *Krox20* MSE, which is methylated at H3K9, and low levels of Krox20 are expressed. Oct6 is also weakly expressed, and KDM3A and low levels of Sox10 and HDAC2 are bound to the *Oct6* SCE HR2, which is methylated at H3K9. In case of lesion, HDAC2, Sox10 and JMJD2C are recruited to the *Oct6* SCE HR2 at 1 dpl to demethylate H3K9 and activate Oct6 expression. Upregulated Oct6 at 1 dpl slows down the upregulation of cJun, which induces the conversion of mature SCs into repair SCs and thereby promotes axonal regrowth. This mechanism can be inhibited by a short-term HDAC1/2 inhibitor treatment, which results in faster conversion into repair SCs and faster axonal regrowth. At 12 dpl, Sox10, HDAC2, KDM3A and JMJD2C are recruited to the *Krox20* MSE to demethylate H3K9 and activate Krox20 expression. Upregulated Krox20 induces remyelination.

using FastStart SYBR Green Master (Roche), according to the manufacturer's recommendations. A dissociation step was added to verify the specificity of the products formed. Primer sequences were as follows: for rat *Oct6*, forward 5′-GGG CACCCTCTACGGTAATG-3′, reverse 5′-CACTTGTTGAGCAGCGGTTT-3′; for rat *Gapdh*, forward 5′-GTATCCGTTGTGGATCTGACAT-3′, reverse 5′-GCCTGC TTCACCACCTTCTTGA-3′, for mouse *Oct6*, forward 5′-AAGCAGTTCAAGCA ACGACG-3′, reverse 5′- CACGTTACCGTAGAGGGTGC-3′, for mouse *Gapdh*, forward 5′- CGTCCCGTAGACAAAATGGT-3′, reverse 5′-TTGATGGCAACAA TCTCCAC-3′.

**Lentiviral production and transduction.** To produce lentiviral particles, HEK293T cells were co-transfected with each lentiviral construct Oct6 shRNA or control shRNA together with the packaging constructs pLP1, pLP2 and pLP/VSVG (Invitrogen) using Lipofectamine 2000 (Invitrogen), according to the recommendations of the manufacturer (ViraPower Lentiviral Expression Systems Manual). Lentiviruses were added to primary RSCs in DM 3 days before adding proliferating medium for dedifferentiation. Rat Oct6 shRNA (TL712899D) and control shRNA (TR30021) lentiviral constructs were purchased from Origene Technologies.

**Semithin and ultrathin sections and electron microscopy.** Mice were killed with 150 mg kg$^{-1}$ pentobarbital i.p. (Esconarkon; Streuli Pharma AG) and sciatic nerves were fixed in situ with 3% paraformaldehyde and 0.15% glutaraldehyde in 0.1 M phosphate buffer, pH 7.4. Fixed tissues were post-fixed in 2% osmium tetroxide, dehydrated through a graded acetone series as described previously[45], and embedded in Spurr's resin (Electron Microscopy Sciences). Semithin sections

were stained with 1% Toluidine blue for analysis at the light microscope, and ultrathin sections (70-nm thick) were made, as described[45]. All analyses were done at 5 mm distal to the lesion site. No contrasting reagent was applied. Images were acquired using a Philips CM 100 BIOTWIN equipped with a Morada side-mounted digital camera (Olympus).

**Chromatin immunoprecipitation.** Sciatic nerves were processed as described[46] (preparation for ChIPseq), with modifications. Briefly, chromatin was fragmented by sonication with a Bioruptor (Diagenode) set at medium power, interval: 30 s on/45 s off, time: 4 cycles of 10 min, rest of 2 min between cycles. Sonicated chromatin containing 75 μg of proteins and 3 μg of antibodies were used per immunoprecipitation (IP). Quantitative real-time PCR analyses were performed with an ABI 7000 Sequence Detection System (Applied Biosystems) using FastStart SYBR Green Master (Roche), according to the manufacturer's recommendations. A dissociation step was added to verify the specificity of the products formed. Primers sequences were as follows: *Oct6* SCE HR2, forward, 5′-CCACTGGGGA GTCCTTCAA-3′, reverse, 5′-TCTCAATGCCAAGGGAGGG-3′, *Krox20* MSE, forward, 5′-TTTCGTCTTTGGGCTCATTC-3′, reverse, 5′-AGCCCTTCACAAAG CTGAAA-3′.

For ChIP carried out with rabbit antibodies, we used rabbit anti-GFP antibody as control IP ( = Neg IP), and for ChIP carried out with mouse antibodies, we used mouse anti-Flag antibody as Neg IP.

**Immunofluorescence.** Mouse sciatic nerves were fixed in situ with 4% PFA for 10 min, dissected, embedded in O.C.T. Compound (VWR chemicals), and frozen at

− 80 °C. Sciatic nerve cryosections (5-µm thick) were first incubated with acetone for 10 min at − 20 °C, washed in PBS/0.1% Tween 20, blocked for 30 min at room temperature (RT) in blocking buffer (0.3% Triton X-100/ 10% Goat serum/ phosphate buffer saline = PBS), and incubated with primary antibodies overnight at 4 °C in blocking buffer. For staining of HDAC1 or HDAC2, cryosections were first incubated with 70% Ethanol for 5 min at RT, washed with PBS and incubated for 40 s with 40 µg ml$^{-1}$ Proteinase K, before incubation with blocking buffer. Sections were then washed three times in blocking buffer and secondary antibodies were incubated for 1 h at RT in the dark. Sections were washed again, incubated with DAPI for 5 min at RT, washed and mounted in Citifluor (Agar Scientific).

For whole-mount staining, nerves were processed as described[47], with the following modifications: sciatic nerves were fixed in 4% paraformaldehyde for 1 h at 4 °C. For blocking and antibody incubations, we used the following buffer: 10% goat serum/1% Triton X-100/PBS.

Primary antibodies: HDAC1 (rabbit, 1:200, Genetex, cat. #GTX100513), HDAC2 (rabbit, 1:100, Santa Cruz Biotechnology, cat. #sc-7899), Oct6 (rabbit, 1:200, kind gift from Dies Meijer), Krox20 (rabbit, 1:100, Axxora, cat. #CO-PRB-236P-100), JMJD2C (rabbit, 1:200, Abcam, cat. #ab85454), KDM3A (rabbit, 1:200, Proteintech, cat. #12835-1-AP), P0 (chicken, 1:500, Aves Labs, cat. #PZO), Neurofilament (chicken, 1:500, GeneTex, cat. #GTX85461; rabbit, 1:200, Millipore, cat. #AB1987), cJun (rabbit, 1:200, Abcam, cat. #ab32137), Sox2 (mouse, 1:100, Millipore, cat. #SC1002), GDNF (rabbit, 1:200, Abcam, cat. #ab18956), GAP-43 (rabbit, 1:500, Abcam, cat. #ab75810). All secondary antibodies were from Jackson ImmunoResearch. Photos were acquired using a Leica TCS SP-II confocal microscope. Single optical sections or z-series projections (stated in the figure legends) are shown. Axonal regrowth was measured using NeuronJ software (ImageJ plugin freely available online with a user manual: http://www.imagescience.org/meijering/software/neuronj/). Parameters used were the same as described in the method validation article[48]: Neurite appearance: Bright, Hessian smoothing scale: 2.0, Cost weight factor: 0.7, Snap window size: 9 × 9, Path-search window size: 2,500 × 2,500, Tracing smoothing range: 5, Tracing subsampling factor: 5, Line width: 1.

**In situ hybridization.** *In situ* hybridization with digoxigenin (DIG)-labelled riboprobes was carried out on cryosections (10-µm thick) as described[49]. Hybridization signals were visualized with NBT/BCIP (Roche Diagnostics). Antisense riboprobes were labelled with digoxigenin according to the manufacturer's instructions (Roche Diagnostics).

**BrdU and EdU assays.** BrDU was injected to mice 2 h prior killing. BrdU assays were carried out as follows: cryosections were post-fixed for 15 min in 4% PFA at RT, washed in PBS, incubated with 2 M HCl for 10 min at 37 °C, rinsed in ddH2O, washed with PBS, digested for 40 s with 40 µg ml$^{-1}$ Proteinase K, washed once with PBS and twice with 0.1% Tween/PBS. Sections were then blocked with 30% goat serum in PBS for 1 h at RT and incubated with BrDU antibody (Roche Diagnostics) in blocking buffer overnight at 4 °C. After three washes with PBS, the secondary antibody (Alexa488 Goat anti-mouse, Jackson ImmunoResearch) was incubated for 1 h in blocking buffer at RT. The sections were washed again in PBS, labelled with DAPI, and mounted in Citifluor. For double-labelling with F4/80 antibody, cryosections were first labelled for F4/80 as described above in the immunofluorescence protocol, then they were processed for BrdU labelling. For EdU assays (Click-iT Plus EdU Imaging Kits, Life Technologies), the EdU was added to primary RSCs 1 h before fixation, and the assay was conducted according to the instructions of the manufacturer.

**Western blot and immunoprecipitation.** For all *in vivo* western blots and immunoprecipitations, we have collected the injured sciatic nerve from the lesion site to around 12 mm distal to the lesion site (region where the nerve splits into the three branches of tibial, sural and common peroneal nerves). We have collected the same region of the contralateral nerve as internal control for each animal. After perineurium removal, sciatic nerves were frozen in liquid nitrogen, pulverized with a chilled mortar and pestle, lysed in radioimmunoprecipitation assay (RIPA) buffer (10 mM Tris/HCl, pH 7.4, 150 mM NaCl, 50 mM NaF, 1 mM NaVO4, 1 mM EDTA, 0.5% wt/vol sodium deoxycholate, and 0.5% Nonidet P-40) for 15 min on ice, and centrifuged to pellet debris. Supernatants were collected, and protein concentration was determined by Lowry Protein assay (Bio-Rad Laboratories).

Cells were washed once in PBS, lysed in RIPA buffer for 15 min on ice, and centrifuged to pellet debris. Sciatic nerves and cell lysates were submitted to SDS-PAGE and analysed by western blotting. Images have been cropped for presentation. Full-size images are presented in Supplementary Figs 17–23.

Primary antibodies used: HDAC1 (rabbit, 1:1,000, Genetex, cat. #GTX100513), HDAC2 (mouse, 1:1,000, Sigma, cat. #H2663), Sox10 (rabbit, 1:250, DCS Innovative Diagnostik-Systeme, cat. #SI058C01), GAPDH (glyceraldehyde-3-phosphate-dehydrogenase, mouse, 1:5,000, Genetex, cat. #GTX28245), P0 (chicken, 1:1,000, Aves Labs, cat. #PZO), Krox20 (rabbit, 1:500, Axxora, cat. #CO-PRB-236P-100), Oct6 (rabbit, 1:2,000, kind gift from Dies Meijer), Sox2 (mouse, 1:250, Millipore, cat. #SC1002), Pax3 (mouse, 1:1,000, Developmental Studies Hybridoma Bank), cJun (mouse, 1:2000, BD Bioscience, cat. #610327), beta-actin (mouse, 1:5,000, Sigma, cat. #A5441), JMJD2C (rabbit, 1:500, Abcam,

cat. #ab85454), KDM3A (mouse, 1:500, Abcam, cat. #ab91252; rabbit, 1:500, Abnova, cat. #PAB16817).

All secondary antibodies were from Jackson ImmunoResearch: light-chain-specific goat anti-mouse-HRP (horse radish peroxidase) and goat anti-rabbit-HRP, and heavy-chain-specific goat anti-chicken-HRP.

For non-denaturing IPs, tissues were prepared and lysed as described in the following RIPA buffer: 10 mM Tris/HCl, pH 7.4, 150 mM NaCl, 50 mM NaF, 100 mM Na3VO4, 1 mM EDTA, 0.5% Na deoxycholate, 0.5% NP40. For denaturing IPs, tissues were lysed in 10 mM Tris/HCl, pH 7.4, 1% SDS, boiled, mixed with nine volumes of RIPA buffer, reboiled, and centrifuged. Lysates were pre-cleared for 1 h with 30 µl protein A/G PLUS agarose beads (Santa Cruz Biotechnology). One millilitre of cleared lysates was rotated overnight at 4 °C with immunoprecipitating antibodies: 2 µg of Sox10 (rabbit, DCS Innovative Diagnostik-Systeme, cat. #SI058C01), KDM3A (mouse, Abcam, cat. #ab91252), JMJD2C (rabbit, Abcam, cat. #ab85454), SUMO-1 (clone D-11 mouse, Santa Cruz biotechnology, cat. #sc-5308), HDAC2 (mouse, Sigma, cat. #H2663), acetyl-lysine (rabbit, Abcam, cat. #ab21623), GFP (rabbit, Abcam, cat. #ab290) or Flag (clone M2 mouse, Sigma, cat. #F1804) antibodies were used per nerve. When IPs were carried out with rabbit antibodies, we used rabbit anti-GFP antibody as control IP. When IPs were carried out with mouse antibodies, we used mouse anti-Flag antibody as control IP. Forty microlitres of beads were added, and samples were rotated for 1 h at 4 °C. Immunoprecipitates were pelleted, washed four times with RIPA buffer, eluted with 15 µl 0.1% formic acid and neutralized with 1.5 M Tris, pH 8.0. Six microlitres of Laemmli buffer were added and samples were boiled for 10 min. Analysis was done by western blot.

**Mass spectrometry analyses.** Mass spectrometry data have been deposited in the ProteomeXchange Consortium via the PRIDE partner repository under accession code PXD005383. Membrane pieces were blocked by incubation in 0.1 M acetic acid with 0.5% polyvinylpyrrolidone-40 (Sigma) for 30 min at 40 °C, washed five times with pure water before protein reduction (50 mM DTT) and alkylation (50 mM iodoacetamide) using 100 µl aliquots for 30 min at 37 °C. Proteins were digested in 15 µl of 10 ng µl$^{-1}$ trypsin solution in 40 mM Tris/HCl pH 8.0 and 10% (v/v) acetonitrile. Digestions were stopped by addition of 10 µl 20% (v/v) TFA.

Proteins from agarose pull-downs were in-solution digested with 100 ng sequencing grade trypsin (Promega) for 6 h at 37 °C after the following treatment: the dry beads were suspended in 30 µl of 8 M urea in 50 mM Tris/HCl, pH 8.0, followed by reduction of the proteins with 3 µl 0.1 M DTT for 30 min at 37 °C and alkylation by addition of 3 µl 0.5 M iodoacetamide for 30 min at 37 °C in the dark, and urea dilution to 2 M by addition of 20 mM Tris/HCl pH 8.0 containing 2 mM CaCl2. Digestion was stopped by adding 1/20 of volume of 20% (v/v) TFA. An aliquot of 5 µl of each digest was analysed by LC-MS/MS on an EASY-nLC1000 chromatograph connected to a QExactive HF mass spectrometer (Thermo Fisher Scientific) using three replicate injections in case of the in-solution digests. Peptides were trapped on a Pepmap100 Trap C18 300 µm × 5 mm (Thermo Fisher Scientific) and separated by backflush onto the analytical column (C18 Aqua Magic, 3 µm, 100 Å, 75 µm × 150 mm) with a 20 or 40 min gradient from 5 to 40% solution B (95% acetonitrile, 0.1% formic acid) at a flow rate of 300 nl min$^{-1}$. Full MS (resolution 60,000, automatic gain control target of 1e6, maximum injection time of 50 ms) and top15 MS/MS (resolution 15,000, target of 1e5, 110 ms) scans were recorded alternatively in the range of 400–1,400 m/z, with an inclusion window of 1.6 m/z, relative collision energy of 27, and dynamic exclusion for 20 s.

Fragment spectra data were converted to mgf with ProteomeDiscoverer 2.0 and peptide identification made with EasyProt software, and processed with MaxQuant/Andromeda version 1.5.0.0 (MQ) searching against the forward and reversed UniprotKB SwissProt mouse protein database (Release 2014_01) with the following parameters: parent mass error tolerance of 10 p.p.m., trypsin cleavage mode with two missed cleavages, static carbamidomethylation on Cys, variable oxidation on Met and acetylation on Lys. On the basis of reversed database peptide spectrum matches, a 1% false discovery rate was set for acceptance of target database matches.

**Data availability.** All relevant data are available from the authors.

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

## Acknowledgements

*P0CreERT2* mice have been used in collaboration with Dr Ueli Suter, floxed *Hdac1/Hdac2* mice with Dr Patrick Matthias and *Oct6* ΔSCE mice with Dr Dies Meijer, who also provided constructs and antibodies. We also thank Drs Pirmin Lötscher, Adolfo Saiardi, Jerry Crabtree, Ed Seto, Stuart Schreiber, Michael Wegner, Piotr Topilko, Toshinobu Nakamura and Toru Nakano for reagents; Drs Thomas Meier, Lester Mills, Ned Mantei and Frank W. Pfrieger for critical reading of the manuscript; and Dr Brian T. Chait for technical advice. Funding was provided by Swiss National Science Foundation Professorship No. PP00P3_139163/ PP00P3_163759, International Foundation for Research in Paraplegia/OPO-Stiftung No. IRP-P147.

## Author contributions

V.B., M.D. and C.J. conceived and designed the experiments. V.B., M.D., M.B., E.M., M.H. and S.R. performed the experiments. V.B., M.D., M.B., E.M., M.H. and C.J. analysed the data. V.B. and C.J. wrote the manuscript.

## Additional information

**Competing financial interests:** The authors declare no competing financial interests.

