## [Peer Review File · Nature Communications]

Reviewers' comments:

Reviewer #1 (Remarks to the Author):

The authors show that schwann cell (SC) genetic and pharmacological HDAC1/2 inhibition promotes peripheral Nerve Regeneration After Lesion by inhibiting SC post-injury redifferentiation. Mechanistically, they show that sumoylated HDAC2 JMJD2 and KDM3A control the redifferentiation program of SC via modulating key genes such as oct6, krox20 and sox10. These data are of interest for both nerve regeneration and mechanisms of SC response after nerve injury.

The manuscript expands upon data from the senior author on SC differentiation and puts it in the context of nerve injury. The authors went a long way to try to provide mechanistic insight into the interesting phenomenon of limiting SC differentiation and myelination to promote nerve regeneration. While the general concept is not absolutely new, the molecular details provided here make this work novel enough to be considered a significant advance.

The experimental design appears sound and the data presented appear mostly convincing.

There are a number of issues that should be addressed to improve the manuscript.

1. The title is not very informative. It must contain the concept at the core of the phenomenon observed that is Schwann Cells dedifferentiation program. I suggest something like: "Epigenetic modulation of Schwann cell redifferentiation program enhances nerve regeneration".
2. The manuscript needs to be edited to make it much more readable. Please introduce briefly the logic behind the choice of each experiment before presenting it, not after, as often the case. For example, why were the specific TF such as oct6 and jun or krox20 chosen for investigation? Why were JMJD2C and KDM3A chosen and not other HME that also affect H3K9me3 and me2/1? I understand this is likely a transferred manuscript from Nature or Nature neuroscience with specific length limitations, but I would spend more time to discuss the data in the context of the literature too.
3. Since sumoylation of HDAC2 post-injury seems key for the initiation of the SC dedifferentiation pathway, can the authors explain what sumoylates HDAC2?
4. It is important that the authors report on the expression of HDAC1 and 2 as well as key competent of the transcriptional complex and target genes in the nerve and inflammatory cells around the injury site. In fact, the expression of the proteins studies is likely not confined to SC and this may affect data interpretation in terms of mechanism.
5. Supplementary figure 7 showing the expression of JMJD2C and KDM3A is not convincing at all. These 2 enzymes do not have to be upregulated to occupy target promoters. I would invite the authors to re-look at their expression level and report on the actual data regardless of whether their expression is up or not after injury.
6. The use of the HDAC1/2 inhibitor does not have any cell specificity, therefore the protein data reported in suppl figure 11 should interpreted with this in mind. The proteins identified are not SC specific.

Minor points:

1. Supplementary Figure 1 and 4 should be moved to the main text and figures as they are rather relevant to the overall mechanistic picture.

Reviewer #2 (Remarks to the Author):

Brügger et al show that HDAC1/2 regulates Oct6 and consequently cJun levels in Schwann cells shortly after injury and Krox20 later. Removal of HDAC1/2 early after injury increases demyelination and consequently axon regeneration at least in part through this pathway. The conclusions that the authors draw are interesting, however the data are very dichotomous. In vitro

studies from primary SCs of complex formation and reporter expression from Oct6 and Krox20 enhancers are convincing. However in vivo phenotype characterization of protein expression, SC dedifferentiation and redifferentiation, regeneration are all very problematic.

Major Issues

The amount and diversity of the work presented is very poorly served by the organization and format. The story is very hard to follow as a result of no introduction, minimal transitions between experiments and such a large amount of data that is central to the story being in supplemental figures. Figures in the main body of the paper are also very complex, often with multiple disjointed foci.

For all in vivo western blots, please specify the location and length of the injured nerve segment that is being analyzed. The quality of the GAPDH band used for loading control is questionable: it appears as doublet or single bands and increase or decrease in injured samples. This GAPDH variability between uninjured and injured samples from the same animal decreases confidence in normalization and quantification by suggesting inherent variability in the system. This is extremely problematic as comparisons are often less than 2-fold change for what appear likely only 3 animals. At least double the N for these experiments to increase confidence and rigor. The HDAC2 (Figure 1A) and Oct6 (Figure 2A) double bands are discussed much later in the paper, which is confusing as to what they are. For HDAC2 and Oct6, it is unclear if both bands quantified and added together. For HDAC2, does the high weight band disappear in the dKO? For HDAC1 (Figure 1A), Co = 1 but shows background levels in the figures, which gives no confidence in that time course. How is compare fold change compared to background? For Oct6 (Figure 2A), why are there 2 bands in Co at 1dpl and 12dpl and not at 3dpl or 5dpl? Or 3 bands in some lanes? Generally, the in vivo western data are too variable and not robust enough (or not representative) to support the quantification as presented.

For 1C, is BrdU increased at baseline in dKO? A similar experiment should be performed at baseline, uninjured conditions in both WT and dKO. A similar question arise with western blots in Figure 2B,C: are there changes in Oct6, cJun in dKO at baseline? To be able to interpret injury response data, knowing baseline expression differences in WT and dKO is required.

Figure 1e is uninterpretable. The method for quantifying axon regeneration is unclear, minimally explained and not cited as experimentally validated. The reference paper for the method used neurofilament staining 5dpl when more degenerating axons have been cleared and the staining is clear. At 3dpl, degenerating/regenerating distinction is much less clear. Also, as SCs have a role in clearing debris and are affected by dKO mutation, axonal degeneration pathway is likely to be affected so axonal markers of degeneration may not be reliable. The images are also too low resolution that colored boxes and arrows show no difference between control and dKO. No scale bars in the colored boxes. A 4-fold change in regeneration rate to almost 4mm per day is very hard to believe. The authors should also verify dKO phenotype with mocetinostat injections.

Minor revisions

Immunofluorescence with "presumptive Schwann cells" needs to be verified with SC marker such as S100b (Figure 1B,C)

For microscopy studies with IF and EM, is the observer blinded for the quantification? If not, should be redone.

Unclear what the point of Figure 1F is. Are there more axons in that image that give rise to increased sprouts? What significance does sprouting have on the phenotype?

Functional recovery N of 3 per data point is too small. Would suggest at least 8 animals per data point due to variability. Also observer for functional recovery must be blinded to treatment.

Reviewer #3 (Remarks to the Author):

The manuscript this referee has seen amounts to a "Results" section only, and Abstract that contains a hint of an introduction and references, an clearly has to be rewritten. This referee has therefore not seen and "Introduction" or a "Discussion" section. These sections have to be reviewed before publication. The following comments apply to the "Results" section only. The key points of the manuscript are the following: (1) By genetically inactivating HDAC1/2 in injured nerves, evidence is provided that the injury-induced upregulation of HDAC1/2 (in particular HDAC2) suppresses the Schwann cell activation that is triggered by injury (measured here by monitoring classical and well established indicators of the reaction of Schwann cells to injury, namely induction of proliferation, myelin breakdown and upregulation of c-Jun). (2) Regeneration is accelerated when HDAC1/2 are inactivated. This is the expected results when signals that suppress the response of Schwann cells to injury are inactivated, because the Schwann cell response to injury promotes regeneration. (3) It was known previously that Oct6 is transiently upregulated in cut or crushed nerves (Monuki et al 1990; Scherer et al 1994), and it is shown here that this activation of Oct6 is driven by HDAC1/2 . The molecular mechanism of the link between HDAC1/2 and Oct6 is analysed in some detail, which identifies additional molecular mechanisms, including JMJD2C and KDM3A and Sox10. (4) It is shown that Oct6 in turn suppresses c-Jun. Because it has been shown elsewhere that c-Jun amplifies the pro-regenerative response of Schwann cells to injury and consequently promotes regeneration (e.g. Jessen and Mirsky 2016), this provides a mechanistic link between HDAC1/2 activation and suppression of regeneration. (5) In line with the finding that HDAC1/2 suppresses c-Jun and regeneration, it is shown that pharmacological inhibition of HDAC1/2 promotes regeneration, an important proof of principle. (6) In addition to activation of Oct6, HDAC1/2 and related mechanisms also promote expression of the pro-myelination transcription factor Krox20. In damaged nerves HDAC1/2 therefore have a double role: (A) They activate Oct6 to suppress c-Jun and therefore dampen the pro-regenerative injury response; (B) they activate Krox20 expression (directly but probably also as a consequence of Oct6 elevation) to promote re-myelination of regenerated axons. (7) As expected from both mechanisms, A and B, remyelination is impaired in the absence of HDAC1/2. These are important data of general interest. Generally the work appears to be technically sound although the quality of many of the western blots is sub-optimal.

Main comments:

(i) I have a major concern over data interpretation. Clearly the link between HDAC1/2 and regeneration and the associated potential for pharmacological intervention is a major strength and interest of this work (rather than the effect of HDAC1/2 on speed of remyelination). This is also acknowledged by the authors in their choice of title (that refers to regeneration). It is therefore unfortunate that the authors take the wrong track in explaining this important pro-regenerative effect. Of the two functions of HDAC1/2 that the authors have so nicely defined (see item (6) above), they opt for mechanism (B) , namely promotion of remyelination as an explanation of accelerated axonal growth (for instance, from the Abstract: "..delaying the induction of the [...] redifferentiation [read :remyelination] program by HDAC1/2 inhibition increases the overall efficiency of peripheral nerve regeneration") There are two problems with choosing this as an explanation of faster regeneration. First there is no evidence in the literature that faster myelination (which only takes place on axons that have regenerated) has any effect on the rate of axon growth. In rodents, re-myelination of an axon is likely to start before it has reached its target. But nobody to my knowledge has shown that the speed of this remyelination has any effect on the rate at which the growing tip of the axon approaches the target site. There is therefore no experimental basis for explaining speed of regeneration by speed of remyelination. Second, there is an obvious alternative and plausible explanation available from the authors own work, namely 6(A) above. The authors should amend the text accordingly

(ii) The authors should strengthen their case still further by measuring the likely elevation of factors such as GDNF that promote regeneration in injured HDAC1/2 inactivated nerves.

(iii) An other aspect of interpretations is of concern. This is the notion that the transient Oct6 expression induced by injury can be read as evidence of redifferentiation, namely myelination. (for instance, from the Abstract: "... the transcriptional program of SC redifferentiation [read: Oct6 activation] starts simultaneously with SC dedifferentiation."). This indicates too simple a view of the role of Oct6. Although a key role of Oct6 is to act as a timer of Krox20 expression in developing nerves, a number of observations show that the presence of Oct6 expression alone cannot be simply interpreted as evidence of myelin differentiation. For instance, Oct6 expression starts in embryonic Schwann cells development long before myelin differentiation (Mandemakers et al. 2000; Blanchard et al 1996). Also, Oct6 expression is induced at the mRNA level at least, albeit not to very high levels, in the distal cut nerves (i.e. not only in crushed nerves) with maximal induction at about day 2 after injury, where clearly myelination is not being initiated (Monuki et al 1990; Scherer et al 1994). Oct6 also potentially represses myelin genes (Monuki et al 1990). Oct6 is also down regulated in cells that have myelinated. Thus Mandemakers et al 2000 discuss the sequence: Oct-6+/Krox-20-, to Oct-6+/Krox-20+ to Oct-6-/Krox-20+. Topilko and Meijer 2001) also discuss a model in which "Oct6 holds ensheathing pre-myelinating cells in an immature stage prior to full myelination" . From all of this it is clear that presence of Oct6 by itself cannot be taken as a sign of remyelination. The text has to be changed accordingly, e.g. lines 26, 27 in Abstract and lines 73,74.

Other comments:

(iv) The induction of proliferation shown in Fig.1C does not discriminate between macrophages that invade nerves at about this time and Schwann cells. It is easy to pin this down to Schwann cells by using a Schwann cell nuclear marker such as Sox 10.

(v) The figure legends, particularly Fig. 2, 3 and 4, are very hard to follow. It would be a great improvement if they were re-organized.

(vi) The notion that Schwann cells convert to a dedifferentiated phenotype (e.g. Abstract line 2) to promote regeneration has been developed further and modified in recent work (see Glen and Talbot 2013; Brosius Lutz and Barres 2014; Jessen and Mirsky 2016). The authors might like to take account of this, since it fits well with their current data.

References quoted in this review:

Brosius Lutz A, Barres BA.

Contrasting the glial response to axon injury in the central and peripheral nervous systems. The repair Schwann cell and its function in regenerating nerves *Dev Cell*. 2014 Jan 13;28(1):7-17. doi: 10.1016/j.devcel.2013.12.002.

Glenn TD, Talbot WS.

Signals regulating myelination in peripheral nerves and the Schwann cell response to injury. *Curr Opin Neurobiol*. 2013 Dec;23(6):1041-8. doi: 10.1016/j.conb.2013.06.010.

Jessen KR, Mirsky R.

The repair Schwann cell and its function in regenerating nerves. *J Physiol*. 2016 Feb 10. doi: 10.1113/JP270874. [Epub ahead of print]

Monuki ES, Kuhn R, Weinmaster G, Trapp BD, Lemke G.

Expression and activity of the POU transcription factor SCIP. *Science*. 1990 Sep 14;249(4974):1300-3.

Scherer SS, Wang DY, Kuhn R, Lemke G, Wrabetz L, Kamholz J.

Axons regulate Schwann cell expression of the POU transcription factor SCIP. *J Neurosci*. 1994 Apr;14(4):1930-42.

Topilko P, Meijer D.

Transcription factors that control Schwann cell development and myelination. In: *Glial Cell development* (Eds KR Jessen and WD Richardson) pp223-244, 2001.

Reviewer #1 (Remarks to the Author):

Opening comment

The authors show that schwann cell (SC) genetic and pharmacological HDAC1/2 inhibition promotes peripheral Nerve Regeneration After Lesion by inhibiting SC post-injury redifferentiation. Mechanistically, they show that sumoylated HDAC2 JMJD2 and KDM3A control the redifferentiation program of SC via modulating key genes such as oct6, krox20 and sox10. These data are of interest for both nerve regeneration and mechanisms of SC response after nerve injury.

The manuscript expands upon data from the senior author on SC differentiation and puts it in the context of nerve injury. The authors went a long way to try to provide mechanistic insight into the interesting phenomenon of limiting SC differentiation and myelination to promote nerve regeneration. While the general concept is not absolutely new, the molecular details provided here make this work novel enough to be considered a significant advance.

The experimental design appears sound and the data presented appear mostly convincing.

There are a number of issues that should be addressed to improve the manuscript.

Response to opening comment

First, we want to thank Reviewer #1 for finding that our work presented in this manuscript provides a significant advance and is of interest for both nerve regeneration and mechanisms of Schwann cell response after nerve injury.

We are also grateful for the comments aimed at improving our manuscript. We find them indeed fully relevant. Here are our answers:

1. The title is not very informative. It must contain the concept at the core of the phenomenon observed that is Schwann Cells dedifferentiation program. I suggest something like: "Epigenetic modulation of Schwann cell redifferentiation program enhances nerve regeneration".

Answer: We agree that a title including the concept of Schwann cell dedifferentiation program or conversion into repair Schwann cells (to combine dedifferentiation process and activation of the repair program; *see Jessen and Mirsky, 2016, doi: 10.1113/JP270874*) would be more informative than the title of our first submission. However, we would prefer not to use the term "epigenetic" because of its controversial definition: some researchers studying epigenetic mechanisms would consider only inherited changes as epigenetics, and dynamic changes as chromatin remodeling. To take into account the comments of Reviewer #3, we would prefer to focus the title on Schwann cell dedifferentiation/conversion into repair Schwann cells

rather than redifferentiation. We have thus renamed our manuscript: “Delaying Histone Deacetylase Response to Injury Accelerates Conversion into Repair Schwann Cells and Nerve Regeneration”

2. The manuscript needs to be edited to make it much more readable. Please introduce briefly the logic behind the choice of each experiment before presenting it, not after, as often the case. For example, why were the specific TF such as oct6 and jun or krox20 chosen for investigation? Why were JMJD2C and KDM3A chosen and not other HME that also affect H3K9me3 and me2/1? I understand this is likely a transferred manuscript from Nature or Nature neuroscience with specific length limitations, but I would spend more time to discuss the data in the context of the literature too.

Answer: We have worked on the text to properly introduce our choice of transcription factors and histone demethylases. Those were chosen by integrating different sources of information: the phenotype of HDAC1/2 dKO nerves analyzed by electron microscopy at different time-points after lesion, the previous literature and the pre-screens we did on the potential involvement of histone demethylases. Indeed, our first submission was formatted for Nature Medicine, and was thus very focused on the actual findings because of length restriction.

3. Since sumoylation of HDAC2 post-injury seems key for the initiation of the SC dedifferentiation pathway, can the authors explain what sumoylates HDAC2?

Answer: In mammalian cells, HDAC2 is known to be sumoylated by small ubiquitin-related modifier (SUMO)-1 (Brandl et al., 2012, *doi:10.1093/jmcb/mjs013*). Sumoylation is achieved by an enzymatic machinery consisting of an E1-activating enzyme (SAE1 or SAE2), the SUMO E2-conjugase Ubc9 and a SUMO E3-ligase (fifteen mammalian enzymes currently known). This machinery catalyzes the covalent attachment of SUMO to lysine residues. Sumoylation is a dynamic posttranslational modification that can be removed by SUMO-specific proteases (SEN1-1 to SEN1-7).

We have added this information to our revised manuscript in the Discussion section on page 18.

4. It is important that the authors report on the expression of HDAC1 and 2 as well as key competent of the transcriptional complex and target genes in the nerve and inflammatory cells around the injury site. In fact, the expression of the proteins studies is likely not confined to SC and this may affect data interpretation in terms of mechanism.

Answer: For all our immunofluorescence analyses of HDAC1, HDAC2, JMJD2C, KDM3A, Oct6, Krox20 and cJun (added in our revised manuscript) on mouse sciatic nerves after lesion, we carried out co-labeling of CD68 or F4/80 to detect expression in macrophages present in the nerves after lesion. In Fig. 1b of our revised manuscript, we have added the co-staining of CD68 with HDAC1 or HDAC2 in contralateral and crushed sciatic nerves, the co-staining of F4/80 with Oct6 in control

and dKO crushed sciatic nerves in Fig. 3e, the co-staining of F4/80 with cJun in Fig. 3g, the co-staining of CD68 with Krox20 in Fig. 5c. In supplementary Fig.10 of our revised manuscript, we have also added the co-staining of F4/80 with JMJD2C and KDM3A. Our data show that while Krox20 is specifically expressed in Schwann cells (as already described), HDAC1, HDAC2, JMJD2C, KDM3A, Oct6 and cJun are also expressed in macrophages at variable levels, but their expression or upregulation in Schwann cells (CD68-negative or F4/80-negative cells with elongated nuclei, typical of Schwann cell nuclei morphology) is robust in sciatic nerves after lesion.

5. Supplementary figure 7 showing the expression of JMJD2C and KDM3A is not convincing at all. These 2 enzymes do not have to be upregulated to occupy target promoters. I would invite the authors to re-look at their expression level and report on the actual data regardless of whether their expression is up or not after injury.

Answer: We agree that JMJD2C and KDM3A do not have to be upregulated to be recruited to target promoters and enhancers. However, while they are both expressed in Schwann cells of adult sciatic nerves at relatively low levels, their expression levels appear higher after lesion (at 5-12dpl). We confirmed this point by Western blot analyses on sciatic nerves lysates at different time-points after lesion. These data are presented in Supplementary Fig. 10 of our revised manuscript. In addition, we have carried out immunofluorescence analyses on primary Schwann cells cultured under differentiating and dedifferentiating conditions. In these cells, we find that KDM3A is upregulated in dedifferentiated Schwann cells and that JMJD2C expression concentrates more in the nucleus and perinuclear region upon dedifferentiation. These additional data are presented in Supplementary Fig.11 of our revised manuscript. However, we have followed the recommendation of Reviewer#1 and have toned down this point and changed the text to the following, on pages 10-11: Compatible with potential functions in regulating Oct6 expression after lesion, JMJD2C and KDM3A were endogenously expressed in SC nuclear and cytoplasmic compartments after nerve lesion (Supplementary Fig. 10a,b) and also in dedifferentiated SCs in culture (Supplementary Fig. 11a,b). Interestingly, they were both upregulated at 5dpl (Supplementary Fig. 10a,b).

6. The use of the HDAC1/2 inhibitor does not have any cell specificity, therefore the protein data reported in suppl figure 11 should be interpreted with this in mind. The proteins identified are not SC specific.

Answer: We agree with this comment. We have added the text below to our revised manuscript, on page 14, where we report the results of the mass spectrometry analyses presented in Supplementary Fig. 14:

HDAC1/2 are ubiquitously expressed, thus HDAC1/2 inhibitor treatment can potentially impact any cell type. A large majority (~85%) of nuclei present in the sciatic nerve at 1dpl belongs to the Schwann cell lineage, while the remaining fraction can be inferred to endoneurial fibroblasts (~10%) and a few macrophages. Therefore, increased abundance of histone peptides and peptides of other nuclear proteins is likely due in large majority to HDAC1/2 inhibition in SCs, however the effect of HDAC1/2 in fibroblast and macrophages may contribute to some extent to the observed results.

Minor points:

7. Supplementary Figure 1 and 4 should be moved to the main text and figures as they are rather relevant to the overall mechanistic picture.

Answer: We have moved most content of Supplementary Figures 1 and 4 of our initial manuscript to the main figures. These data are now presented in Fig. 3 and 4 of our revised manuscript.

Reviewer #2 (Remarks to the Author):

Opening comment

Brügger et al show that HDAC1/2 regulates Oct6 and consequently cJun levels in Schwann cells shortly after injury and Krox20 later. Removal of HDAC1/2 early after injury increases demyelination and consequently axon regeneration at least in part through this pathway. The conclusions that the authors draw are interesting, however the data are very dichotomous. In vitro studies from primary SCs of complex formation and reporter expression from Oct6 and Krox20 enhancers are convincing. However in vivo phenotype characterization of protein expression, SC dedifferentiation and redifferentiation, regeneration are all very problematic.

Response to opening comment

We thank Reviewer #2 for finding the conclusions of our work interesting. We have worked on improving the robustness of our data. We hope that the additional work we have carried out to revise our manuscript renders our data convincing and strengthens our conclusions.

Here are our point-by-point answers to the specific comments:

Major Issues

1. The amount and diversity of the work presented is very poorly served by the organization and format. The story is very hard to follow as a result of no introduction, minimal transitions between experiments and such a large amount of data that is central to the story being in supplemental figures. Figures in the main body of the paper are also very complex, often with multiple disjointed foci.

Answer: we have re-organized our manuscript, wrote an introduction, explained in more details the transitions between experiments, transferred some of the data formerly presented as supplementary material to the main body of the manuscript, and split the data previously presented in four main figures into ten main figures in our revised manuscript. We think that our manuscript is now improved and easier to follow with this re-organization.

2. For all in vivo western blots, please specify the location and length of the injured nerve segment that is being analyzed.

Answer: for all in vivo Western blots, we have collected the injured sciatic nerve from the lesion site to around 12 mm distal to the lesion site (region where the nerve splits into the three branches of tibial, sural and common peroneal nerves). We have collected the same region of the contralateral nerve as internal control for each animal. We have added this information in the Methods section, in the Western blot and immunoprecipitation subsection.

3. The quality of the GAPDH band used for loading control is questionable: it appears as doublet or single bands and increase or decrease in injured samples. This GAPDH variability between uninjured and injured samples from the same animal decreases confidence in normalization and quantification by suggesting inherent variability in the system. This is extremely problematic as comparisons are often less than 2-fold change for what appear likely only 3 animals. At least double the N for these experiments to increase confidence and rigor.

Answer: we indeed often detect a double band of 35-37 kDa for GAPDH signal by Western blot. While GAPDH is known to be phosphorylated, we think that the double band we detect is most likely due to sample processing rather than phosphorylation, because we could not identify a correlation of this double band with any particular condition or time-point. We have searched the literature to find out whether other articles show this double band, and indeed we found several examples. To cite two: Guzhova et al., *Human Molecular Genetics*, 2011, doi:10.1093/hmg/ddr314: Fig. 1B; Shang-Hua et al., *Journal of Biomedical Science*, 2009, doi:10.1186/1423-0127-16-40: Fig. 2A-B. In these two studies, the authors mentioned this double band of GAPDH as possible result of sample processing: “It was also noted that monomer GAPDH was presented on western as a double 35–37 kDa band. It is possible that the recruitment of GAPDH into SDS-insoluble aggregates triggers an excision of GAPDH, thus producing a lower molecular weight fragment; alternatively, this fragment can be a result of partial degradation of the enzyme or an oxidation of certain sulphhydryl groups.”, and “Please note that there was a low molecular band, recognized by anti-GAPDH specific antibody, shown only in the nuclear fraction, which may be a derived and/or spliced product from the intact GAPDH.”

In all our Western blots, we have quantified the double band detected with GAPDH antibody. To make sure that this signal can be used as loading control, we have blotted a membrane with GAPDH antibody, and reblotted the same membrane with beta-actin antibody. On this membrane, six cell lysates from two different experiments were loaded. The relative quantification of GAPDH signal is very similar to the relative quantification of beta-actin signal (see results presented below). Thus, we are confident that the double band detected with the GAPDH antibody can be used as reliable loading control.

The same membrane was blotted with GAPDH antibody, and re-blotted with beta-actin antibody

Quantification by ImageJ

GAPDH				
	Area	Mean	Min	Max
1	0.043	137.101	43	229
2	0.043	98.154	42	213
3	0.043	147.725	44	221
1'	0.043	114.074	44	220
2'	0.043	124.932	43	208
3'	0.043	89.576	43	213

Quantification by ImageJ

b-actin				
	Area	Mean	Min	Max
1	0.025	130.234	21	231
2	0.025	96.354	20	206
3	0.025	159.007	25	230
1'	0.025	141.58	24	223
2'	0.025	146.893	21	228
3'	0.025	108.226	21	218

GAPDH			ratio compared to #1 or 1'
1	137.101		1
2	98.154		0.715924756
3	147.725		1.077490317
1'	114.074		1
2'	124.932		1.095183828
3'	89.576		0.785244666
b-actin			ratio compared to #1 or 1'
1	130.234		1
2	96.354		0.73985288
3	159.007		1.22093309
1'	141.58		1
2'	146.893		1.037526487
3'	108.226		0.764415878

We have however increased the N number of our in vivo Western blots. For the in vivo Western blots presented in our initial manuscript, we had used 3 to 6 animals per time-points. Indeed, the Veterinary office of the Canton of Fribourg recommends that we use the minimal number of animals allowing for significant results. However, in our revised manuscript, we have repeated the time-points where we had 3 or 4 animals to increase the N number to 5 or 6 per time-point (we used the maximum number of animals of the right age we had available). With this increased N number, we confirm the conclusions of our initial manuscript. We are convinced of the robustness of our data and we hope that the N number is now appropriate.

4. The HDAC2 (Figure 1A) and Oct6 (Figure 2A) double bands are discussed much later in the paper, which is confusing as to what they are.

Answer: In our revised manuscript, we have moved the description of the two bands for HDAC2 and Oct6 just after reporting on their upregulation after lesion.

5. For HDAC2 and Oct6, it is unclear if both bands quantified and added together.

Answer: For both HDAC2 and Oct6, we have quantified each band and added them together in our revised manuscript. It was not the case in our initial submission, where we had quantified only the lower molecular weight band of HDAC2 and Oct6. We have added this information in the legend of Fig. 1 for HDAC2 and of Fig. 3 for Oct6.

6. For HDAC2, does the high weight band disappear in the dKO?

Answer: In the dKO, both bands (high and lower molecular weight) detected by the HDAC2 antibody disappear. See below at 3dpl. We have added this information in our revised manuscript page 6, Supplementary Fig. 1.

7. For HDAC1 (Figure 1A), Co = 1 but shows background levels in the figures, which gives no confidence in that time course. How is compare fold change compared to background?

Answer: We have quantified HDAC1 levels with blots longer exposed where we could detect low levels of HDAC1 in the contralateral nerve lysate. To avoid confusion on this point, we have replaced the photos of our initial manuscript by photos where we can see HDAC1 in the contralateral nerve lysates, however at very low levels most of the time. These very low levels are still higher than background levels. We did the same for cJun, Pax3 and Sox2.

8. For Oct6 (Figure 2A), why are there 2 bands in Co at 1dpl and 12dpl and not at 3dpl or 5dpl? Or 3 bands in some lanes? Generally, the in vivo western data are too variable and not robust enough (or not representative) to support the quantification as presented.

Answer: In contralateral nerves, we always detect a double band at 50-52kDa and a single band at 42 kDa, at any time-point after lesion. The double band appears weaker at 3dpl and 5dpl in Fig. 2A of our initial submission because the photos presented are taken from a low exposed film, but at higher exposure the double band is visible. The different time-points after lesion were not run at the same time, therefore signal intensity between time-points is not directly comparable. To make this point clearer, we have replaced the photos with longer exposures where the double band at 50-52kDa is visible. It is true that we initially paid more attention to the lower molecular weight band that is upregulated at all time-points after lesion, whereas the higher molecular weight band is only upregulated at 1dpl.

9. For 1C, is BrdU increased at baseline in dKO? A similar experiment should be performed at baseline, uninjured conditions in both WT and dKO.

Answer: The percentage of BrDU-positive cells is not affected in contralateral nerves of dKO compared to contralateral nerves of Control, at any time-point after lesion. In fact, we did not detect BrDU-positive cells in either dKO or Control contralateral nerves. We have added this information on page 6, Supplementary Fig.2a.

10. A similar question arises with western blots in Figure 2B,C: are there changes in Oct6, cJun in dKO at baseline? To be able to interpret injury response data, knowing baseline expression differences in WT and dKO is required.

Answer: In contralateral nerves of dKO, the levels of cJun and Oct6 are not affected compared to Control contralateral nerves, except for a moderate decrease of Oct6 at 12 days post lesion. We have added these data on pages 7-8, Supplementary Fig.5.

11. Figure 1e is uninterpretable. The method for quantifying axon regeneration is unclear, minimally explained and not cited as experimentally validated. The reference paper for the method used neurofilament staining 5dpl when more degenerating axons have been cleared and the staining is clear. At 3dpl, degenerating/regenerating distinction is much less clear. Also, as SCs have a role in clearing debris and are affected by dKO mutation, axonal degeneration pathway is likely to be affected so axonal markers of degeneration may not be reliable. The images are also too low resolution that colored boxes and arrows show no difference between control and dKO. No scale bars in the colored boxes. A 4-fold change in regeneration rate to almost 4mm per day is very hard to believe. The authors should also verify dKO phenotype with mocetinostat injections.

Answer: To quantify axonal outgrowth, we have used NeuronJ software, which is a plugin for ImageJ that has been developed by Eric Meijering et al. (2004, doi: 10.1002/cyto.a.20022). This is a freely available online software (<http://www.imagescience.org/meijering/software/neuronj/>) developed specifically to trace neurites and axons. The manual to use the software can be found at the following link: <http://www.imagescience.org/meijering/software/neuronj/manual/>. For quantification, we have used the default NeuronJ parameters, which are the ones used in the method validation paper cited above (Meijering et al., 2004). Parameters used were as follows: Neurite appearance: Bright, Hessian smoothing scale: 2.0, Cost weight factor: 0.7, Snap window size: 9x9, Path-search window size: 2500x2500, Tracing smoothing range: 5, Tracing subsampling factor: 5, Line width: 1.

We have added this information and the validation method reference to the Methods and the References section.

Reference: E. Meijering, M. Jacob, J.-C. F. Sarria, P. Steiner, H. Hirling, M. Unser. Design and Validation of a Tool for Neurite Tracing and Analysis in Fluorescence Microscopy Images. *Cytometry Part A*, vol. 58, no. 2, April 2004, pp. 167-176

We have indeed tried several other time-points to quantify axonal regrowth: 3, 5 and 7 days post lesion. To decide on the best time-point to use for whole-nerve staining, we

made 20- μ m thick cryosections of control and dKO nerves at these time-points and stained them with Neurofilament. We found that at 5 days and 7 days post lesion, dKO nerves had already regrown a lot. The 3-day post lesion time-point appeared ideal, because we could see axonal regrowth in dKO nerves, but less than at 5 and 7 days post lesion, whereas control nerves showed very little axonal regrowth (See images of 3 and 7dpl presented below). We thus decided to use 3 days post lesion for whole-nerve imaging, because with cryosections it is not possible to image the whole axonal length. At 3 days post lesion, we can already differentiate between regenerated and degenerated axons. Regenerated axons display a bright Neurofilament staining that is not interrupted, whereas degenerated axons show a staining that is either bright or at background level, but is fragmented.

To do a whole-nerve staining of Neurofilament, we used the protocol of David Parkinson's group (published last year), which worked very nicely. We have referenced this article in our initial submission already.

According to the comments of Reviewer#2 on markers of axonal degeneration that may not be reliable at 3dpl, we have removed SMI32 staining from Figure 1e.

We have used higher resolution photos in our revised manuscript, and we have added scale bars to the colored boxes.

We have also carried out whole-nerve neurofilament immunofluorescence analyses on sciatic nerves of mice treated with Mocetinostat or vehicle for 3 days after lesion. These new results show faster axonal regrowth in Mocetinostat-treated animals compared to vehicle, consistently with our data on dKO nerves. We have added these results on page 14, Supplementary Fig. 15.

Minor revisions

12. Immunofluorescence with "presumptive Schwann cells" needs to be verified with SC marker such as S100b (Figure 1B,C)

Answer: We have tried different protocols and antibodies to double-stain adult crushed nerves with HDAC1 or HDAC2 and S100b. We found a good protocol to stain HDAC1 and HDAC2 using rabbit antibodies, but unfortunately we could not make work any mouse antibodies (including antibodies for HDAC1, HDAC2 or S100b), because of the high background after lesion, most likely due to mouse immunoglobulins. To reduce this background, we have tried protocols with Fab fragments as blocking agents, but those were not good enough to obtain a convincing staining. Thus, we tried double-stainings with a goat S100b antibody, but this did not work well either. Thus we decided to double-stain for HDAC1 or HDAC2 and F4/80 or CD68 as markers of macrophages. These stainings with rat primary antibodies and secondary antibodies adsorbed against mouse immunoglobulins (minimal cross-reaction with mouse) gave good results and allowed us to differentiate between macrophages and Schwann cells. Apart from these two cell types, there is also some contribution from fibroblasts, but they represent about 1/10 of the number of Schwann cells. In addition, Schwann cell nuclei display a typical elongated shape, which make them easy to recognize.

To strengthen our point that HDAC1 and HDAC2 are upregulated in Schwann cells after lesion, we have added co-labelings of HDAC1 or HDAC2 with CD68 after lesion. These data replace the previous stainings in Fig. 1B.

We also carried out double-labeling of BrDU with the macrophage marker F4/80. We show in our revised manuscript (page 6, Supplementary Fig. 2b) that all BrDU-labeled cells at 3dpl are negative for F4/80, strongly suggesting that the BrDU-positive cells detected at 3dpl are Schwann cells.

13. For microscopy studies with IF and EM, is the observer blinded for the quantification? If not, should be redone.

Answer: We have re-labeled the photos with only the animal numbers without information on their genotype or treatment, and have re-done the quantifications blinded. The new data confirm the previous analyses.

Those correspond to the data presented in our initial submission in the following figures:

- For IF: Fig. 1e (revised: Fig. 2c), Fig. 2f (revised: Fig. 4b), Fig. S4d (revised: Fig. 4f)
- For EM: Fig. 1f (revised: Fig. 2d), Fig. 1g (revised: Fig. 2e), Fig. 4h (revised: Fig. 9d), Fig. S4c (revised: Fig. 4e)

Concerning Fig. 1c (IF) and Fig. 1d (EM) of our initial submission (revised: Fig. 2a and Fig. 2b, respectively), quantifications were already done blinded by a naïve experimenter (Master student, who was not involved in this project at this time and did not know which results would be expected).

14. Unclear what the point of Figure 1F is. Are there more axons in that image that give rise to increased sprouts? What significance does sprouting have on the phenotype?

Answer: At 5 days post lesion, it is difficult to differentiate between sprouting that will give rise to axons reconnecting to their target or to sprouting that will eventually degenerate. This is why we counted any axon that had regrown. We used the term axonal sprouts to count axonal outgrowth. We would not know how to confidently identify sprouts arising from the same axon. We could reconstruct a piece of nerve by electron microscopy images using the FIBS technique; however, this is to our knowledge doable for a short tissue length. It would thus be extremely challenging to reconstruct a 5-mm piece of adult sciatic nerve. In any case, the point with this piece of data was not to state on the number of sprouts per axon. What we want to show with these data is that axons regrow more efficiently in the absence of HDAC1/2, whether those will eventually reconnect or not to their peripheral target for functional recovery. To count the number of axons that have successfully regrown and that may lead to functional recovery, we would have to count them at 1 month post lesion, when axons are remyelinated or when they have reassembled in Remak bundles. This is what we have done with the Mocetinostat treatments to show the improvement of axonal regeneration after treatment. We did not do this analysis in dKO nerves, because remyelination is impaired in the absence of HDAC1/2. This defect may influence axonal survival, and even if this is not the case, the interest of an increased number of axons that are unmyelinated or poorly remyelinated is not obvious.

15. Functional recovery N of 3 per data point is too small. Would suggest at least 8 animals per data point due to variability. Also observer for functional recovery must be blinded to treatment.

Answer: We have increased the N to 8 or 9 per data point for functional recovery experiments. The observer was not the person who operated and treated the mice and had never seen the mice before carrying out the behavioral tests. The observer only received the animal number as information. The animal number is given at birth by the animal caretaker. No information on whether the animals received vehicle, 3-day Mocetinostat or 5-day Mocetinostat treatment was given to the observer.

Increasing the number of animals per data point also increased the significance of our results and thus strengthened our conclusions. We are therefore thankful to Reviewer#2 for this suggestion.

Reviewer #3 (Remarks to the Author):

Opening comment

The manuscript this referee has seen amounts to a "Results" section only, and Abstract that contains a hint of an introduction and references, an clearly has to be rewritten. This referee has therefore not seen and "Introduction" or a "Discussion" section. These sections have to be reviewed before publication. The following comments apply to the "Results" section only.

The key points of the manuscript are the following: (1) By genetically inactivating HDAC1/2 in injured nerves, evidence is provided that the injury-induced upregulation of HDAC1/2 (in particular HDAC2) suppresses the Schwann cell activation that is triggered by injury (measured here by monitoring classical and well established indicators of the reaction of Schwann cells to injury, namely induction of proliferation, myelin breakdown and upregulation of c-Jun). (2) Regeneration is accelerated when HDAC1/2 are inactivated. This is the expected results when signals that suppress the response of Schwann cells to injury are inactivated, because the Schwann cell response to injury promotes regeneration. (3) It was known previously that Oct6 is transiently upregulated in cut or crushed nerves (Monuki et al 1990; Scherer et al 1994), and it is shown here that this activation of Oct6 is driven by HDAC1/2 . The molecular mechanism of the link between HDAC1/2 and Oct6 is analysed in

some detail, which identifies additional molecular mechanisms, including JMJD2C and KDM3A and Sox10. (4) It is shown that Oct6 in turn suppresses c-Jun. Because it has been shown elsewhere that c-Jun amplifies the pro-regenerative response of Schwann cells to injury and consequently promotes regeneration (e.g. Jessen and Mirsky 2016), this provides a mechanistic link between HDAC1/2 activation and suppression of regeneration. (5) In line with the finding that HDAC1/2 suppresses c-Jun and regeneration, it is shown that pharmacological inhibition of HDAC1/2 promotes regeneration, an important proof of principle. (6) In addition to activation of Oct6, HDAC1/2 and related mechanisms also promote expression of the pro-myelination transcription factor Krox20. In damaged nerves HDAC1/2 therefore have a double role: (A) They activate Oct6 to suppress c-Jun and therefore dampen the pro-regenerative injury response; (B) they activate Krox20 expression (directly but probably also as a consequence of Oct6 elevation) to promote re-myelination of regenerated axons. (7) As expected from both mechanisms, A and B, remyelination is impaired in the absence of HDAC1/2.

These are important data of general interest. Generally the work appears to be technically sound although the quality of many of the western blots is sub-optimal.

Response to opening comment

We first want to thank Reviewer#3 for a perfect outline of the key points of our study and for finding that our work provides important data of general interest. We are also thankful for the comments aimed at a more accurate interpretation of our data. We agree with these comments and we have made changes in our revised manuscript accordingly. We have also discussed our current data in relation to the literature cited by Reviewer#3.

Here are below our answers to the specific comments.

Main comments:

- (i) I have a major concern over data interpretation. Clearly the link between HDAC1/2 and regeneration and the associated potential for pharmacological intervention is a major strength and interest of this work (rather than the effect of HDAC1/2 on speed of remyelination). This is also acknowledged by the authors in their choice of title (that refers to regeneration). It is therefore unfortunate that the authors take the wrong track in explaining this important pro-regenerative

effect. Of the two functions of HDAC1/2 that the authors have so nicely defined (see item (6) above), they opt for mechanism (B) , namely promotion of remyelination as an explanation of accelerated axonal growth (for instance, from the Abstract: "...delaying the induction of the [...] redifferentiation [read :remyelination] program by HDAC1/2 inhibition increases the overall efficiency of peripheral nerve regeneration"). There are two problems with choosing this as an explanation of faster regeneration. First there is no evidence in the literature that faster myelination (which only takes place on axons that have regenerated) has any effect on the rate of axon growth. In rodents, re-myelination of an axon is likely to start before it has reached its target. But nobody to my knowledge has shown that the speed of this remyelination has any effect on the rate at which the growing tip of the axon approaches the target site. There is therefore no experimental basis for explaining speed of regeneration by speed of remyelination. Second, there is an obvious alternative and plausible explanation available from the authors own work, namely 6(A) above. The authors should amend the text accordingly.

Answer: We actually totally agree with Reviewer#3 about the interpretation of our data. We are convinced that the increased speed of axonal regeneration in the absence of HDAC1/2 (or when HDAC1/2 activity is inhibited) is due to earlier activation of the injury pro-regenerative response driven by cJun, and not by delayed remyelination. Because Oct6 described function is to induce Schwann cell differentiation and re-differentiation, we made the interpretation that the early redifferentiation program driven by Oct6, but not the remyelination driven by Krox20, slows down the dedifferentiation program driven by cJun. We also show that Oct6 upregulation after injury is induced by a transcriptional complex that includes Sox10, a major inducer of Schwann cell differentiation and re-differentiation. This is another reason why we thought that the inhibitory effect of Oct6 on cJun upregulation is part of the re-differentiation program. However, it is not excluded that this early function of Oct6 in slowing down cJun upregulation is not part of the re-differentiation program, but is a distinct function of Oct6. To avoid confusion on this point, we have changed the text, as requested by Reviewer#3.

(ii) The authors should strengthen their case still further by measuring the likely elevation of factors such as GDNF that promote regeneration in injured HDAC1/2 inactivated nerves.

Answer: We have carried out immunofluorescence analyses of GDNF in Control and dKO sciatic nerves at 1, 3, and 5 days post lesion, when there is a likely elevation of this factor in dKO nerves compared to Controls, due to earlier and higher cJun upregulation. Indeed, our stainings confirm this expected increase of GDNF in the absence of HDAC1/2, by detectable GDNF expression already at 3 days post lesion in dKO nerves, whereas GDNF was detectable only at 5 days post lesion in Control nerves. We have added these data on page 8, Supplementary Fig. 6. We thank Reviewer#3 for this suggestion.

(iii) An other aspect of interpretations is of concern. This is the notion that the transient Oct6 expression induced by injury can be read as evidence of

redifferentiation, namely myelination. (for instance, from the Abstract:"... the transcriptional program of SC redifferentiation [read: Oct6 activation] starts simultaneously with SC dedifferentiation."). This indicates too simple a view of the role of Oct6. Although a key role of Oct6 is to act as a timer of Krox20 expression in developing nerves, a number of observations show that the presence of Oct6 expression alone cannot be simply interpreted as evidence of myelin differentiation. For instance, Oct6 expression starts in embryonic Schwann cells development long before myelin differentiation (Mandemakers et al. 2000; Blanchard et al 1996). Also, Oct6 expression is induced at the mRNA level at least, albeit not to very high levels, in the distal cut nerves (i.e. not only in crushed nerves) with maximal induction at about day 2 after injury, where clearly myelination is not being initiated (Monuki et al 1990; Scherer et al 1994). Oct6 also potentially represses myelin genes (Monuki et al 1990). Oct6 is also down regulated in cells that have myelinated. Thus Mandemakers et al 2000 discuss the sequence: Oct - 6+/Krox - 20-, to Oct - 6+/Krox - 20+ to Oct - 6-/Krox - 20+. Topilko and Meijer 2001) also discuss a model in which "Oct6 holds ensheathing pre-myelinating cells in an immature stage prior to full myelination" . From all of this it is clear that presence of Oct6 by itself cannot be taken as a sign of remyelination. The text has to be changed accordingly, e.g. lines 26, 27 in Abstract and lines 73,74.

Answer: As mentioned in our answer to comment (i), we agree with Reviewer#3 that Oct6 may have other functions not related to the re-differentiation program, and we have thus changed the text accordingly, as requested.

Other comments:

(iv) The induction of proliferation shown in Fig.1C does not discriminate between macrophages that invade nerves at about this time and Schwann cells. It is easy to pin this down to Schwann cells by using a Schwann cell nuclear marker such as Sox 10.

Answer: We have actually tried various protocols and several antibodies to stain Sox10 in adult mouse sciatic nerves, but without success.

As explained in our answer to comment 12 of Reviewer#2, we have however double-labeled BrDU with the macrophage marker F4/80. We show in our revised manuscript (page 6, Supplementary Fig. 2b) that all BrDU-labeled cells at 3dpl are negative for F4/80, strongly suggesting that the BrDU-positive cells detected at 3dpl are Schwann cells.

(v) The figure legends, particularly Fig. 2, 3 and 4, are very hard to follow. It would be a great improvement if they were re-organized.

Answer: We have re-organized the figure legends and split the four figures presented in our initial submission into ten figures. We hope that the figure legends are now simpler to follow.

- (vi) The notion that Schwann cells convert to a dedifferentiated phenotype (e.g. Abstract line 2) to promote regeneration has been developed further and modified in recent work (see Glen and Talbot 2013; Brosius Lutz and Barres 2014; Jessen and Mirsky 2016). The authors might like to take account of this, since it fits well with their current data.

References quoted in this review:

Brosius Lutz A, Barres BA.

Contrasting the glial response to axon injury in the central and peripheral nervous systems. The repair Schwann cell and its function in regenerating nerves. *Dev Cell*. 2014 Jan 13;28(1):7-17. doi: 10.1016/j.devcel.2013.12.002.

Glenn TD, Talbot WS.

Signals regulating myelination in peripheral nerves and the Schwann cell response to injury. *Curr Opin Neurobiol*. 2013 Dec;23(6):1041-8. doi: 10.1016/j.conb.2013.06.010.

Jessen KR, Mirsky R.

The repair Schwann cell and its function in regenerating nerves. *J Physiol*. 2016 Feb 10. doi: 10.1113/JP270874. [Epub ahead of print]

Monuki ES, Kuhn R, Weinmaster G, Trapp BD, Lemke G.

Expression and activity of the POU transcription factor SCIP. *Science*. 1990 Sep 14;249(4974):1300-3.

Scherer SS, Wang DY, Kuhn R, Lemke G, Wrabetz L, Kamholz J.

Axons regulate Schwann cell expression of the POU transcription factor SCIP. *J Neurosci*. 1994 Apr;14(4):1930-42.

Topilko P, Meijer D.

Transcription factors that control Schwann cell development and myelination. In: *Glial Cell development* (Eds KR Jessen and WD Richardson) pp223-244, 2001.

Answer: We thank Reviewer#3 for this suggestion and for providing the references of the articles. Indeed, in our revised article, we have discussed our data in relation to the work referenced by Reviewer#3, and cited this work.

Reviewers' comments:

Reviewer #1 (Remarks to the Author):

The authors have addressed my comments and improved the manuscript.

I remain unhappy about supp figure 10 where the quality of the immunofluorescence is poor. Plus, there is no quantification of the expression of the proteins shown in the blot, bar graphs need to be reported with N of at least 3.

Overall, as presently shown, the interpretation of data of this figure is not supported by the evidence.

Otherwise, the manuscript is worth publishing in my view as it reports on a novel important piece of biology with mechanistic insight overall well supported by the data.

Reviewer #2 (Remarks to the Author):

Brügger et al submitted a greatly improved manuscript and addressed criticisms from all reviewers. The amount of work that has gone into this paper is well-appreciated. To that end, the data regarding biochemical interactions of transcription factors and their effects on gene expression after sciatic nerve lesion is convincing and consistent with the immunostaining and EM showing the effects on Schwann cells. However, some concerns remain.

Overall the quality of western blot presented has improved. However a more detailed discussion in the results section would be helpful, especially when bands are present as doublet. For example, in their response to reviewers the authors now say they quantify both bands for Oct6. However, Oct6 lower and upper bands after injury (Figure 3) and during development (Supp figure 4) behave in the exact opposite manner. This render data confusing as to what is quantified and whether Oct 6 increases or decreases. A discussion about these two isoforms and what their role might be would be useful.

The axon regeneration data remains somehow problematic, even with the addition of the experiment in which mice were treated with Mocetinostat or vehicle for 3 days after lesion. Although this reviewer appreciates the effort put towards quantifying axon regeneration in the whole nerve with whole mount neurofilament staining, the magnitude of the increased axon growth in the dKO mice is still very hard to believe with the image quality provided, especially considering this is a cell non-autonomous effect on axon growth. This is in part due to a lack of precedence to see axons grow that much faster in vivo as well as the images submitted being visually unclear to this reviewer. The neurofilament staining does not appear to match the pink tracing and the reason why different colored box along the different parts of the nerve are presented is not explained in the result section, rendering the data confusing. Additionally, the high magnitude of growth is not well correlated with a similar magnitude increase in functional recovery. While the 3 day treatment shows "faster improvement" in toe pinch at early time points, it's not clear why functional recovery occurs in one toe within 10 days, and then another 8 days to get the next two. Together these inconsistencies don't sit well. The authors may strongly consider removing the axon growth data; the functional recovery data presented in Figure 9 is sufficient to prove their point. Or at an absolute minimum a second method of axon growth determination would be needed to increase confidence.

Reviewer #3 (Remarks to the Author):

The authors have added an Introduction that clearly describes the background to the work described in the Results section. They have also embellished the discussion of their results that now takes care of all the main points that need to be addressed.

They have also fully addressed all the points raised in my previous review including the addition of new experiments, technical revision of Western blots, and better organization and clarity of data

presentation.

Overall this interesting manuscript is greatly improved.

Reviewer #1 (Remarks to the Author):

The authors have addressed my comments and improved the manuscript. I remain unhappy about supp figure 10 where the quality of the immunofluorescence is poor. Plus, there is no quantification of the expression of the proteins shown in the blot, bar graphs need to be reported with N of at least 3. Overall, as presently shown, the interpretation of data of this figure is not supported by the evidence.

Otherwise, the manuscript is worth publishing in my view as it reports on a novel important piece of biology with mechanistic insight overall well supported by the data.

Answer: We thank Reviewer #1 for the positive comments about the importance of our work. We are happy that we could address the comments appropriately and that our manuscript is now improved.

For Supplementary Fig. 10, we have quantified our Western blots (N=3 animals per time-point) and presented the data in bar graphs. These data show that KDM3A and JMJD2C are both significantly upregulated at 5 days post lesion. Because the immunofluorescence data did not appear convincing, we decided to present a high magnification photo of KDM3A or JMJD2C and F4/80 (macrophage marker) co-immunofluorescence at 5 days post lesion to better show that they are both expressed in Schwann cells and macrophages. Note that KDM3A and JMJD2C are expressed in cytoplasmic and nuclear compartments of Schwann cells. This point is strengthened by Supplementary Fig. 11 where we show indeed cytoplasmic and nuclear localization of KDM3A and JMJD2C in cultured dedifferentiated primary Schwann cells.

Reviewer #2 (Remarks to the Author):

Brügger et al submitted a greatly improved manuscript and addressed criticisms from all reviewers. The amount of work that has gone into this paper is well-appreciated. To that end, the data regarding biochemical interactions of transcription factors and their effects on gene expression after sciatic nerve lesion is convincing and consistent with the immunostaining and EM showing the effects on Schwann cells. However, some concerns remain.

Overall the quality of western blot presented has improved. However a more detailed discussion in the results section would be helpful, especially when bands are present as doublet. For example, in their response to reviewers the authors now say they quantify both bands for Oct6. However, Oct6 lower and upper bands after injury (Figure 3) and during development (Supp figure 4) behave in the exact opposite manner. This render data confusing as to what is quantified and whether Oct 6 increases or decreases. A discussion about these two isoforms and what their role might be would be useful.

Answer: We thank Reviewer #2 for acknowledging the improvement of our manuscript and for appreciating the amount of work we have dedicated to this study. We are also happy that our data are convincing.

Concerning Oct6, we have further explained in the Results section the different regulation of each band to avoid confusion and we have discussed the possible identity of these high and low molecular weight bands. This clarification is written on pages 7 and 8.

The axon regeneration data remains somehow problematic, even with the addition of the experiment in which mice were treated with Mocetinostat or vehicle for 3 days after lesion. Although this reviewer appreciates the effort put towards quantifying axon regeneration in the whole nerve with whole mount neurofilament staining, the magnitude of the increased axon growth in the dKO mice is still very hard to believe with the image quality provided, especially considering this is a cell non-autonomous effect on axon growth. This is in part due to a lack of precedence to see axons grow that much faster in vivo as well as the images submitted being visually unclear to this reviewer. The neurofilament staining does not appear to match the pink tracing and the reason why different colored box along the different parts of the nerve are presented is not explained in the result section, rendering the data confusing. Additionally, the high magnitude of growth is not well correlated with a similar magnitude increase in functional recovery. While the 3 day treatment shows “faster improvement” in toe pinch at early time points, it’s not clear why functional recovery occurs in one toe within 10 days, and then another 8 days to get the next two. Together these inconsistencies don’t sit well. The authors may strongly consider removing the axon growth data; the functional recovery data presented in Figure 9 is sufficient to prove their point. Or at an absolute minimum a second method of axon growth determination would be needed to increase confidence.

Answer:

In the Results section of our revised manuscript, on page 6, we have explained the use of the colored boxes. The magenta tracing does match the Neurofilament staining. We carried on tracings until the staining signal stops, we did not trace fragmented axons. We trace at high magnification where we can follow the path of axons.

Axonal regrowth in Mocetinostat-treated animals is 2 to 3 times faster than in vehicle at 3 days post lesion. This faster axonal regrowth correlates well with faster functional recovery, however it seems difficult to directly correlate axonal regrowth at 3 days post lesion with the progress of functional recovery made between 10 and 18 days post lesion. It is likely that the speed of axonal regrowth is not as fast at a later time-point. Indeed, Mocetinostat treatment is only given during the first 3 days following lesion. And even in the dKO, the increase of cJun expression compared to controls lasts for the first 2 days post lesion, likely acting as a pulse for axonal regrowth. In addition, between 10 and 18 days post lesion, remyelination may also play a role in functional recovery (some sensory fibers are myelinated). We would like to keep the axon growth data by whole-nerve immunofluorescence. We like this technique because it allows measuring the actual length of axons from the lesion site. To increase confidence in our claim that axons regrow faster in the absence of HDAC1/2, we have added immunofluorescence data on cryosections of dKO and control nerves to show an additional marker of axonal regrowth. We chose GAP-43 that is well described to be upregulated in regrowing axons. These data show a strong increase of GAP-43 expression in dKO nerves compared to control nerves at 3dpl and 5dpl. We

have added these data in Figure 2d of our revised manuscript. In addition to immunofluorescence data, we show by electron microscopy that more axons have regrown in dKO compared to control nerves: we found a comparable increase of axonal regrowth by electron microscopy as we found by immunofluorescence. We are thus very confident with our axonal regrowth data, which we hope are now convincing.

Reviewer #3 (Remarks to the Author):

The authors have added an Introduction that clearly describes the background to the work described in the Results section. They have also embellished the discussion of their results that now takes care of all the main points that need to be addressed. They have also fully addressed all the points raised in my previous review including the addition of new experiments, technical revision of Western blots, and better organization and clarity of data presentation.

Overall this interesting manuscript is greatly improved.

Answer: We thank reviewer #3 for finding our manuscript interesting and greatly improved. We are happy that we could appropriately revise it.

REVIEWERS' COMMENTS:

Reviewer #2 (Remarks to the Author):

The authors have addressed my comments and the additional data and detailed discussion of results improved clarity of this very interesting manuscript.